

# The importance of process interactions and parameter sensitivity for modelling the carbon dynamics in a natural peatland

Christine Metzger[1*], Mats B. Nilsson[2], Matthias Peichl[2] and Per-Erik Jansson[1]

[1]Department of Land and Water Resources Engineering, Royal Institute of Technology, Stockholm, 100 44, Sweden
[2]Department of Forest Ecology & Management, Swedish University of Agricultural Sciences, Umeå, 901 83, Sweden
[*]now at: Institute for Meteorology and Climate Research/Atmospheric Environmental Research (IMK-IFU), Karlsruhe
Institute for Technology, Garmisch-Partenkirchen, 82467, Germany

*Correspondence to*: Christine Metzger (cmetzger@kth.se)

**Abstract.** In contrast to previous peatland carbon dioxide ($CO_2$) model sensitivity analyses, usually focusing on only one or

few modules, this study investigates interactions between various biotic and abiotic processes and their parameters by comparing CoupModel results with multiple observation variables.

Many interactions were found not only within, but also between the various model modules. Each measurement variable was sensitive to up to ten (out of 54) parameters, from up to seven different processes. The constrained parameter ranges varied, depending on the variable and performance index chosen as criteria, and on other calibrated parameters (equifinalities).

Therefore, transferring parameter ranges between models needs to be done with caution, especially if such ranges were achieved by considering only few processes. The identified interactions and constrained parameters will be of high interest to use for comparisons with model results and data from similar ecosystems. All of the available measurement variables (net ecosystem exchange, leaf area index, sensible and latent heat fluxes, net radiation, soil temperatures, water table depth and snow depth) improved model constraint. Additional measurements of soil hydraulic properties or water content would reduce

equifinalities and constrain additional parameters that showed high range of uncertainty.

The presented results can help modellers and experimentalists to design model calibrations and experimental setups on peatlands.

**Keywords.** Parameter uncertainty, equifinalities, net ecosystem exchange (NEE), carbon dioxide ($CO_2$), boreal mire

## 1 Introduction

Understanding and quantification of interactions between different processes and between different parameters is required for reducing uncertainty in prognostic modelling in carbon (C) cycle research. Undisturbed peatlands act as carbon sink and have accumulated at least 550 Gt of C, which is equivalent to twice the C stock in the forest biomass of the world (Gorham, 1991;

Parish, 2008). A more recent estimate for exclusively northern peatlands amounts to 436 Gt of C (Loisel et al., 2014).



Management or climate change can cause this carbon to be released as $CO_2$ emissions (Maljanen et al., 2010; Drösler et al., 2013; Petrescu et al., 2015). Process oriented models are necessary to transfer the knowledge gained from measurements to different locations, management or future climate scenarios. Further, such models can help to understand the processes underlying the observations. Only few of the parameters used in process models are known as site independent, unambiguous

constants from laboratory experiments. All others need to be either assumed, or gained from calibration procedures (e.g. Kennedy and O'Hagan, 2001, Wang and Chen 2012). But not all parameters have a strong impact on model output and performance (i.e. fit between modelled and measured variables). Monte Carlo based sensitivity analysis are used to identify key parameters for both, the performance and the impact on various major model outputs (e.g. Verbeeck et al., 2006; Van Oijen et al., 2011; Santaren et al., 2014).

Many studies investigated single processes and their parameters, while only few consider different biotic and abiotic processes and multiple calibration variables. Several modelling studies have explored peatland hydrology (e.g. Dimitrov et al., 2010; Dettmann et al., 2014) and heat fluxes in peatlands (e.g. Granberg et al., 1999; Keller et al., 2004), while others concentrate on carbon fluxes and pools (e.g. Frolking et al., 2002; Verbeeck et al., 2006; Wu et al., 2013) where the focus is sometimes on heterotrophic respiration only (e.g. Abdalla et al., 2014).

Models are continuously extended or coupled with other models (e.g. Wang et al., 2005; Prentice et al., 2007; Giltrap et al., 2009; Hidy et al., 2012; Jansson 2012; Tang et al., 2015), developing to more and more holistic models, accounting for plant and soil carbon processes, water and energy flows and biochemistry. However, often only parameters of the new module are tested (e.g. Belassen et al., 2010; Wania et al., 2010, Zhu et al., 2014; Tang et al., 2015), ignoring possible interactions between modules.

Many peatland studies investigate only the sensitivity to carbon fluxes or pools, despite their models include different biotic and abiotic processes (e.g. Yurova et al., 2007; St. Hilarire et al., 2010; Quillet et al., 2013; Webster et al., 2013; Wu and Blodau, 2013; Kim et al., 2014). Though, the profit of using multiple constraints for model calibration and the importance of interactions between parameters and across modules has been shown by e.g. sensitivity analyses on forest ecosystems (Carvalhais et al., 2010; Santaren et al., 2014; Tian et al., 2014).

Further, often only local sensitivity analyses are performed, changing only one parameter or one input variable at a time (e.g. Hilbert et al., 2000; Yu et al., 2001; Zhang et al., 2002; Wania et al., 2009; Frolking et al., 2010; Tang et al., 2010; St-Hilaire et al., 2010). This approach does not account for possible interactions and non-linearity in equations (e.g. Saltelli et al., 2008; Quillet et al., 2013), but peatland processes are often non-linear and interact in many ways (Belyea, 2009).

Net ecosystem exchange is the balance of photosynthesis, and respiration from plants and microbes. All three NEE components

are strongly interconnected in several ways with the amount of plant biomass, temperature, radiation, nutrients and moisture availability (e.g. Clymo, 1984; Lindroth et al., 2007). Photosynthesis, soil temperature (Ts) and moisture depend among others on incoming radiation, transpiration and plant coverage. Heterotrophic respiration further depends on quality and quantity of plant litter (e.g. Yeloff and Mauquoy, 2006). In addition, phenological events such as the timing of snow melt are important for soil temperature dynamics, biologic activity and peatland $CO_2$ fluxes (Aurela, 2004; Peichl et al., 2015).





Such processes interactions are realised in complex ecosystem models, but lead to inter-correlation between the different parameters and complicate the parameter constraint to an unambiguous solution: several combinations of different parameter values can lead to a similar good fit of model output to measured variables which is defined as equifinality (Beven and Freer, 2001). The model sensitivity to such parameters might be hidden if equifinalities are not considered. Constraining a model

based on multiple variables can help to resolve or reduce equifinalities (Carvalhais et al., 2010). Unlike previous peatland modelling studies, we therefore investigate the sensitivity to parameters from several different modules simultaneously, in their effect on not only on NEE, but also on LE, sensible heat (H), net radiation (Rn), leaf area index (LAI), Ts, WT and snow, and identify parameter interactions.

However, criteria based on multiple variables imply a subjective weighting of variables and performance indices. Fitting the

model to a certain variable might improve or worsen the performance in another variable (Carvalhais et al., 2010) and might therefore have implications for the parameter range judged as valid (e.g. Schulz and Beven, 2003). In this study, the effects of selecting a certain criteria on the resulting parameter range will be investigated.

We use the detailed ecosystem model CoupModel (Jansson and Karlberg, 2010) for the following reasons: It is a well-established and widely used model (Jansson, 2012). Its model structure is flexible and allows simulation of different abiotic

and biotic processes based on well-established physical equations, which can be selected by the user. The CoupModel includes all main components expected to have an impact on the carbon cycle: i) A detailed module for simulation of heat and water fluxes in the soil and at the interface to the atmosphere, ii) plant growth from photosynthesis, limited by water availability and temperature, iii) plant respiration and litter fall and iv) a module for soil organic carbon (SOC) decomposition. A user defined time step allows using the full information contained in measurements with high temporal resolution (i.e. hourly) on site scale.

**1.1 Objectives**

The aim was to identify and explore the connections within and between biotic and abiotic processes and parameters which are relevant for modelling NEE in a natural open peatland, by investigating several different output variables. The specific objectives were:

1. To identify the processes and parameters that have the strongest impact on model performance

2. To evaluate implications of different criteria selection choices on model performance and resulting parameter ranges

3. To identify and describe equifinalities between parameters from different processes

4. To test the usability of all available observation data for model constrain and identify missing measurement variables

The answers to these questions will be crucial for future model calibrations on similar ecosystems: They will represent most valuable information for selecting processes that need to be taken into account, for selecting parameters and their value ranges

and considering parameter connections, as well as selecting sites and observed variables. They further help experimentalists to decide on the measurement of variables to make their site suitable for modelling.



## 2 Materials and methods

### 2.1 Site description

Degerö Stormyr (64.182016 N, 19.55663 E) is an oligotrophic, minerogenic, mire, located on a highland, 270 m.a.s.l, in the county of Västerbotten, Sweden. A detailed description of the site and the measurements can be found in Peichl et al. (2014)

and references therein. "The mire catchment is predominantly drained by the small creek Vargstugbäcken towards north-west. The depth of the peat is generally between 3–4 m, but depths up to 8 m have been measured. … The micro-topography is dominated by mainly carpets and lawns, with only sparse occurrences of hummocks" (Peichl et al., 2014). The plant community of the mire is dominated by cottongrass (*Eriophorum vaginatum* L), tufted bulrush (*Trichophorum cespitosum* L. Hartm.) and *Sphagnum* mosses (Nilsson et al., 2008; Laine et al., 2012). Total aboveground biomass (moss capitula and

vascular plants) is $141 \pm 45$ g m$^{-2}$ (Laine et al., 2012). Seasonal maximum leaf area index of vascular plants was estimated at 0.8 m$^2$ m$^{-2}$ in 2012 (Peichl et al., 2015).

The 30-year (1961–1990) mean annual precipitation and air temperature are 523 mm and $+1.2°C$, respectively, while the mean air temperatures in July and January are $+14.7°C$ and $-12.4°C$, respectively (Alexandersson et al., 1991). The snow cover normally reaches a depth of up to 0.6 m and lasts for approximately 6 months (Peichl et al., 2014). The peatland was

continuously a sink for atmospheric $CO_2$ during twelve years of Eddy covariance measurements, with a 12-year average ($\pm$ standard deviation) NEE of $-58 \pm 21$ g C m$^{-2}$ yr$^{-1}$ (Peichl et al., 2014).

### 2.2 Data used in this study

Hourly values of global radiation, air temperature, relative humidity, precipitation and wind speed were used as meteorological

input data (Table 1). They were measured at the same tower as the EC sensors. An overview of the data used for calibration can be found in Table 2; a more detailed description in Peichl et al. (2014) and references therein. For gap filling (due to instrument failure) as well as for the pre-evaluation period 1991-2000, daily data from the nearby (13 km away) standard climate station at the Svartberget field station were obtained. In case of air temperature and relative humidity, seasonal regression relationships were applied to account for temperature and humidity differences between the site and the standard

climate station.

Measured carbon concentrations per soil layer were used for estimation of pool sizes as described in Sect. 2.3.5. The model was calibrated based on measured NEE, LE, H, WT, Rn, soil temperatures in $-2$ cm (Ts$_1$) and $-42$ cm (Ts$_2$) depth, snow depth and LAI of vascular plants as listed in Table 2, and described in Peichl et al. (2014), and references therein. Negative NEE values indicate net $CO_2$ uptake while positive NEE values indicate emission of $CO_2$. All calibration data were averaged to

hourly values, except LAI values and snow depth. In this study, only measured values were used for calibration: gap-filled values during measurement gaps were omitted.



## 2.3 Model description and application to Degerö Stormyr

CoupModel v5 from 12[th] December 2014 was used for simulations. The current version can be downloaded from the CoupModel homepage (CoupModel, 2015). A detailed description can be found in Jansson and Karlberg (2010). The model represents the ecosystem by a description of C and N fluxes in the soil and in the plants. It includes the main abiotic fluxes, such as soil heat and water fluxes that represent the major drivers for regulation of the biological components of the ecosystem. For application to Degerö Stormyr, the vegetation canopy was defined as two layers: vascular plants and mosses. The soil profile was divided into 16 layers with an increasing layer depth from 4 cm for the upper nine layers to 60 cm in the lowest layer, resulting in a total depth of 3.4 m. The model internal time step was half-hourly for abiotic processes and hourly for nitrogen and carbon related processes. The simulations were started ten years prior to the evaluation period, so the system could adapt to the site conditions and become more independent of initial values.

The most important equations and the corresponding calibrated parameters can be found in Table S1 and S2 in the supplement. The major model assumptions relating to the model application to the peatland are described below. Detailed assumptions in respect to fixed parameter values can be found in Table S3 in the supplement.

### 2.3.1 Radiation interception, evapotranspiration and snow

An interception model for both, radiation and precipitation, a snow model and a surface pool model was used to provide boundary conditions at the soil surface. Cloud fraction was calculated from global radiation input and latitude. Incoming radiation was partitioned between one part, which was absorbed by the plant canopy and another part, which reached the soil according Beer's law (cf. Impens and Lemeur, 1969). Radiation absorbed by the canopy was partitioned between the two plant layers (Fig. 1), depending on their height and surface cover, whereas it was assumed that leaves are uniformly distributed within the total height of the canopy. Interception and plant evaporation depended on the simulated leaf area index of the vegetation as well as the degree of area coverage. Transpiration depended additionally on the simulated plant water uptake. Soil evaporation was derived from an iterative solution of the soil surface energy balance of the soil surface, using an empirical parameter for estimating the vapour pressure and temperature at the soil surface. Vapour pressure deficit was calculated from the relative humidity input. Snow fall was simulated from precipitation and air temperature, while snow melt was estimated from global radiation, air temperature and simulated soil heat flux.

### 2.3.2 Soil temperatures and heat fluxes

Surface temperature was simulated based on an energy balance approach, where the radiation reaching the soil equals the sum of sensible and latent heat flux to the air and heat flux to the soil. The same approach was used for the snow surface temperature. Heat flow between adjacent soil layers were calculated based on thermal conductivity functions accounting for the content of





ice and water. The heat flow equation is based on a coupled equation also accounting for the freezing and thawing in the soil (Jansson and Halldin, 1979). Convection heat flows were not accounted for. The lower boundary temperature was calculated based on a sine variation including parameters for the annual mean temperature and amplitude at the site.

### 2.3.3 Soil hydrology

Soil water flows and water contents were calculated for each of the 16 soil layers. Soil water depended on infiltration to the soil, soil evaporation, water uptake by plants, and ground water flow. Soil moisture represented as liquid water content, is calculated based on the water storage and temperature in the corresponding soil layer. Water flows between adjacent soil layers were calculated based on Richards' equation (Richards, 1931), considering hydraulic conductivity, water potential gradient

and vapour diffusion. Saturation conductivity was assigned depending on the mean measured dry bulk density values of the corresponding layers (cf. Päivänen, 1973).

In respect to hydrologic characteristics, the soil profile was divided in two horizons representing the acrotelm and the catotelm (cf. Ivanov, 1981), whereas the boundary between these horizons was positioned at −30 cm as suggested for Degerö Stormyr, based on visual differences in the soil profile and water table depth measurements (Granberg et al., 1999). The soil water

characteristics were described by the Brooks & Corey equation (Brooks and Corey, 1964) and unsaturated conductivity by the Mualem function (Mualem, 1976). When the current simulated ground water table is above the assumed drainage level, outflow of saturated layers above that level was simulated, based on a linear model.

Surface runoff was controlled by a surface pool of water that covers various fractions of the soil surface. During periods of a fully saturated soil profile the flow of water in the upper soil compartment could be directed up-wards, towards the surface

pool. Surface runoff was calculated as a function of the amount of water in the surface pool.

### 2.3.4 Vegetation

Two plant layers were simulated, representing vascular plants and mosses. They differed in their parameters for size, shape, C allocation, litter fall and temperature response for assimilation and respiration.

Vascular plants consisted of three functional parts: roots, photosynthetically active biomass (i.e. green leaves and green stems that are labelled as leaves in equations and parameter names), and photosynthetically passive biomass (i.e. brown, senescent leaves and woody stems that are labelled as stems). Mosses were considered to consist of two parts: an upper, photosynthetically active part (labelled as leaves) and a lower, photosynthetically passive part (labelled as roots) representing pale or brown, belowground leaves and stems that are still living. Each plant constitutes a biomass pool for each of its parts.

Vascular plants had additionally a pool for mobile reserves. LAI was proportional to leaf biomass by using a constant specific leaf area as conversion factor. Vascular plants were assumed to have a maximal height of 50 cm compared to 2 cm for mosses.



Plant development started every spring when the accumulated sum of air temperatures above a threshold value reached a certain value. The accumulation of temperatures started when the day length (geometric estimated time of sun above horizon) exceeded 10 hours. Snow cover hindered leafing-out by reducing the radiation supply to the plant, while low soil temperatures reduced plant water uptake.

Senescence and litter fall differed between the two plant types. For vascular plants, beside a small amount of litter fall occurring during the whole plant growth period (cf. Fulkerson and Donaghy, 2001), senescence was assumed to start after the plant reached maturity and therefore depended on growth stage (cf. Thomas and Stoddart, 1980) and dormancy temperatures (cf. Davidson and Campbell, 1983). New assimilates were constantly allocated to the roots and to the photosynthetically active part. After maturity, existing green biomass was reallocated to the photosynthetically passive part. A third stage of litter fall
was configured depending on a temperature threshold: Five consecutive days in the autumn with day lengths shorter than 10 hours and with temperatures below a threshold temperature parameter terminated the growing season; Increased litter fall took place and vascular plants went to dormancy. During vascular plant litter fall, part of the carbon was stored in the mobile pool, which could be then reused for leafing-out in the next year (cf. White, 1973; Wingler, 2005). The litter from above ground biomass was inserted to a surface litter pool, while root litter was inserted to the corresponding litter pools of the soil layers in
which the roots were located. The litter in the surface pool was inactive and transferred with a constant rate to the litter pool of the uppermost layer.

A different approach for senescence and litter fall was applied for mosses, as they largely differ in these processes from vascular plants: *Sphagnum* mosses produce new leaves in the top (capitula), while litter fall occurs on the lower leaves, when they become shaded and die (cf. Clymo and Hayward, 1982). This leads to a permanent moss cover and a litter fall that is
proportional to assimilation. In the model, this was realised by keeping the photosynthetically active part of mosses to a fixed static value. Any losses (i.e. respiration and litter fall) or gains (incorporation of assimilates) were restricted to the belowground moss parts. Moss litter was produced with a constant rate coefficient throughout the year and was directly inserted to the corresponding soil litter pools. The dormancy period for mosses was initiated in the same way as for vascular plants, but affected only assimilation.

For both plant types, assimilation was simulated using the light use efficiency approach (cf. Monteith, 1972), at which total plant growth is proportional to the net of global radiation absorbed by the canopy but limited by unfavourable temperature and limited soil water. The response to soil water was defined from the ratio of actual to potential transpiration. Potential transpiration depended on vapour pressure, temperature, wind speed and aerodynamic resistance of the plant. Actual transpiration was assumed to equal water uptake from soil layers, depending on relative amount of roots, the specific response
to soil water potential, and soil temperature of each layer. Both plant layers were assumed to be well adapted to wet conditions (cf. Keddy, 1992; Steed et al., 2002) and therefore experiencing water stress only due to too dry conditions, which was supported by pre-study modelling results.

Plant respiration was assumed to be proportional to assimilation (growth respiration) and to amount of biomass (maintenance respiration) in active leaves and roots. In case of mosses, maintenance respiration took place only in belowground parts,



therefore a higher range for the parameter scaling growth respiration was calibrated (cf. Table S1 in the supplement). A simple Q10 approach was used to simulate the response of plant maintenance respiration on temperature.

### 2.3.5 SOC decomposition

The organic substrate was represented by three C and N pools for each of the 16 soil layers: one representing more stable, partly decomposed material ($SOM_s$), one representing fresh or little decomposed moss litter ($SOM_m$) and one representing fresh or little decomposed litter from vascular plants ($SOM_v$). Initial conditions were selected to fulfil the measured total carbon per layer and partitioned into the pools in the way that they were approximately in equilibrium for a certain parameter combination that produces a reasonable fit to NEE (prior calibration).Decomposition products from the $SOM_m$ and $SOM_v$ pools were

partitioned into $CO_2$ which was released to the atmosphere and C which is partly moved to the $SOM_s$ pools and partly returned to the $SOM_m$ and $SOM_v$ pools. Decomposition products from the $SOM_s$ pools were partly released as $CO_2$ and partly returned to the $SOM_s$ pools. Under saturated conditions, carbon could leave the pools as methane ($CH_4$), which was later oxidised to $CO_2$ or transported to the atmosphere via plants or through ebullition. The rate at which carbon was transferred between pools and towards the atmosphere was pool specific and reduced under unfavourable soil temperature and moisture conditions.

Temperature dependence was described with a function which was developed by Ratkowsky et al. (1982) for bacteria, but has also been applied to fungal growth (Bazin and Prosser, 1988). The response to moisture was assumed to be zero at moisture contents below the wilting point, rising to 100% between two threshold moisture contents and falling to a certain level under saturated conditions.

Peat depth growth during the simulation period was considered by the following: The initial organic concentration was

preserved for each layer but the lowest in the profile. Instead, the difference in the total amount of C in all pools in one layer between start and end of each year was moved to or from the layer below, to simulate growth or decrease of the peat depth. Thereby, carbon was taken from the different pools according to the relative abundance of each pool in the source layer and inserted to the corresponding pool in the target layer to allow dynamic changes in litter quality. The lowest layer (−2.8 to −3.4 m below the surface) represented the entire depth change of the whole profile, but was excluded from a constant concentration

to avoid adjustments of the number of layers.

Nitrogen and methane related processes were considered by a model including the most important pathways and fluxes. However no emphasize on the calibration of these processes were made in this study.

### 2.4 Calibration procedure

A Monte Carlo calibration including acceptance criteria was performed and the resulting parameterisations were analysed for correlations between different parameters, between parameters and model performance and between performances in different



variables, to identify process and parameter interactions. 50'000 runs were performed, using a uniform random distribution within assumed prior ranges for 54 selected parameters from different modules. The parameters were selected as candidates to demonstrate the role of various regulating processes. Many parameters were still considered with fixed single values (Table S3 in the supplement). Prior ranges for calibrated parameters were selected according to literature values or experiences from

previous model runs, in most cases a certain range around the default values (Table S1 in the supplement). Model outputs were compared with measured field data including many variables in high temporal resolution, spanning up to 12 years of observations (Table 2). Several combined criteria were defined to select runs (behavioural models) with an acceptable performance (see Sect. 2.4.2) in different variables. Resulting parameter value ranges of the accepted runs were then compared with the prior ranges and between the different criteria selections to examine the effect of criteria selection. Correlations

between parameter values and model performance in the different measurement variables were analysed, as well as between accepted values of different parameters. Parameters were ranked in their effect on model performance, their correlation with other parameters and their constrain ability from the available data.

### 2.4.1 Splitting of calibration variables into sub-periods

Additional to the calibration data for the whole period we introduced further sub-variables for certain sub periods and times of the day. NEE was separated into night time values (22:30 – 02:30), representing ecosystem respiration, and day time values (09:30 – 15:30), representing the sum of the respiration component and the assimilation component. Additionally, spring time values were considered separately for NEE and snow depth, and spring and winter time values for Rn, Ts, H, and LE. This is justified as low values with little dynamic during winter and the critical transition of plant emerge and snow melt in spring

might not be properly accounted for, if only the whole period was considered. WT was calibrated and analysed in the whole profile and additional in lower soil layers (one sub-variable for WT depths $> -0.15$ m and one for $> -0.2$ m). This was motivated, as WT in the upper soil layers showed high fluctuations in the modelled, and also partly the measured WT, while our interest was to achieve a good overall water table with good representations of dry summer periods.

### 2.4.2 Performance indices

Selection of runs and evaluation of model performance were based on three indices: coefficient of determination ($R^2$) asses how well the dynamics in the measurement derived values are represented by the model. Mean error (ME) is the difference between the average of the simulated compared to the average in the measured, i.e. it shows the

error in the magnitude. Nash-Sutcliff efficiency (NSE) (Nash and Sutcliffe, 1970) accounts for both, deviation of dynamics and magnitude. It ranges from $-\infty$ to 1, whereas 1 means the best fit of modelled to measured data. Values $< 0$ indicate that



the mean measured value is a better predictor than the simulated value (Moriasi et al., 2007). As NSE may be understood as a combination of $R^2$ and ME, it was only evaluated, if $R^2$ and ME alone did not narrow the parameter range.

NEE showed a spiky record, especially during night time, probably caused by transport processes in the atmosphere, which were not represented in the model. To attenuate the effect of the spikes, the simulated and measured values were transformed

to cumulated total amounts, starting from the beginning of the observation period. An additional $R^2$ value was calculated for the cumulated values ($AR^2$).

### 2.4.3 Criteria for posterior selection

Criteria were applied in two steps. In the first step, a basic set of 1285 behavioural models was selected. Out of these, several sets of 50 runs each were selected in the second step in two different ways: one for sensitivity analyses and parameter ranges

which was based on single criteria and the other for identification of equifinalities, based on multiple criteria.

**Basic selection**

The basic selection was applied, as the lowest summer water levels and a reasonable representation of the plant was assumed to be crucial for most of the processes of interest. Criteria were on performance in WT and vascular plant LAI (Table 3). The

criteria on water level below 0.2 m was chosen, as a correct representation of summer drought conditions was of higher interest in this study than a correct water level during e.g. frozen conditions in winter, causing water table drop downs to 0.15 m. The criteria on LAI ME of $\pm 0.2$ m$^2$ m$^{-2}$ was a relatively wide range, as the mean of measured values was 0.4 m$^2$ m$^{-2}$, i.e. a underestimation of LAI by $-0.2$ m$^2$ m$^{-2}$ would result in a maximum LAI of 0.2–0.4, which was close to the minimum for being able to re-establish new biomass after a low productive year. A wide range of day-time NEE ME was additionally applied to

exclude outliers due to numerical problems, which reached an ME in NEE up to $8 \cdot 10^{27}$ gCO$_2$-C day$^{-1}$ m$^{-2}$ in the prior.

**Single criteria to identify parameter range**

For sensitivity analyses and to test if, and how, parameter ranges depend on the selected criteria, the best 50 behavioural models for each performance index of each variable were selected out of the basic selection. Thereby, best means highest in case of

$R^2$ and NSE, but closest to zero in case of ME. We defined posterior parameter range as the interval between the 5% and the 95% percentile of the distribution of parameter values of the runs selected. Posterior parameter ranges were compared with the ranges resulting from the basic selection. If the upper or lower limit of a posterior parameter range of the final selections differed by $\geq 10\%$ from the upper or lower limit of the posterior range of the basic selection, the parameter was assumed to be sensitive to the selected criteria and further analysed.

The same was done for each best 200 behavioural models, but as the results were similar, they were only plotted in respect to parameter ranges. Further, all parameters were plotted against all performance indices of each variable and checked visually for discrepancies with the resulting ranges (results are not shown).





**Multiple criteria to identify parameter correlations**

For identification of equifinalities, a set of multiple criteria for each variable (Table 3) was applied to select sets of 50 behavioural models each. Again, these selections were based on the basic selection. Parameter ensembles of these accepted behavioural models were then analysed to identify covariance between parameters. A pair of parameters was considered to

interact, if their values correlated with an $R^2$ of at least 0.1 in the basic selection, respectively 0.2 in the final selection. If a pair showed correlations in several criteria sets, the highest $R^2$ value was reported in the results.

### 2.4.4 Evaluation and measures

To rank the parameters in their concern, several measures were used to quantify parameter sensitivities and constrain-abilities, as well as equifinalities. The sensitivity ($S$) of a parameter to each performance index of each variable was quantified by the

sum of the differences between posterior range and prior range (range reduction). If a parameter was sensitive to more than one period of each variable, the highest value for each variable was chosen for further analysis. To identify trade-offs and supporting effects between different criteria, correlations of the performances between different variables and indices were plotted and visually analysed. Due to limited computer capacity, this was based on a random set of 3200 runs. Further, the parameter value ranges resulting from the different criteria were compared with each other and determined how well they were

overlapping, i.e. how unambiguously they could be constraint. Overlap ($O$) for each parameter was defined as the difference between the minimum of the upper limits of the posterior ranges of the different criteria, minus the maximum of the lower limits of posterior ranges and therefore become negative, if ranges were not overlapping. Further it was compared how well overlapping ranges differed between performance indices within the same variable and between different variables. The overlapping range of each parameter was normalized by dividing it by the average of the posterior ranges of this parameter, so

that a value of 1 would be reached if all posterior ranges of that parameter would be identical for all performance indices and variables. Equifinalities were quantified by the $R^2$ value of the correlation between each parameter pair. Parameter concern ($P$) was defined based on three components: the sensitivity of the parameter, how unambiguously it could be constraint and the sum of correlation coefficients of equifinalities with other parameters:

$$P = (S_{R^2} + S_{ME}) \times (1 - O) + \sum 2 \times \frac{2^{10 \times R^2_{equi}}}{10} \tag{1}$$

Thereby, sensitivity was the sum of the range reduction for $R^2$ and for ME, respectively NSE in case no sensitivity was detected for $R^2$ and ME but NSE. The sensitivity was multiplied by the factor one minus the normalized overlapping range, so that the sensitivity of parameters which could be unambiguously constrained are down weighted, and such with high uncertainty due to different results for different performance indices or variables are up weighted. Equifinalities were considered by the sum of $R^2$ values for each correlation of that parameter with another parameter, displayed in exponential form and weighted, so that

strong correlations were emphasised and the contribution of equifinalities were in a comparable scale to the sensitivity measures.





## 3 Results

Processes as well as parameters were strongly interacting, which was reflected in sensitivities of each variable to several different modules, correlations between the performance in different variables, and in equifinalities between parameters of different modules.

About half of the parameters were sensitive to model performance in one or more variables, but only very few had a distinct range (Sect. 3.1). Instead they affected several processes, causing trade-offs in model performance between the different measurement variables, but also several supporting effects could be identified (Sect. 3.2). A lot of equifinalities were identified between parameters. Parameters were correlated with up to seven other parameters, often from different modules. Therefore, a good performance often requires certain combinations of parameter values, rather than specific parameter values (Sect. 3.3).

Each of the available measurement variables (NEE, LAI, sensible and latent heat fluxes, net radiation, soil temperatures, water table depth and snow depth) constrained several parameters, without any variable being redundant (Sect. 3.4). Nevertheless, large uncertainty remained in especially the unsaturated water distribution ($\psi_a$) in the soil (Fig. 2), which affected all considered processes and hindered further parameter constrain. This might be solved by additional measurements of i.e. soil hydraulic properties. Other important parameters that could not be constrained, define aerodynamic resistance, radiation interception (in

particular moss albedo), timing of snow melt, and in case of NEE mostly the leaf litter fall rate of vascular plants during the growing season (Fig. 2).

A detailed description of the key parameters for each process and the detected interactions can be found in Sect. 3.5. Results for model fits to the different variables can be found in Fig. S1 in the supplement.

### 3.1 Parameter sensitivity

Most of the 27 sensitive parameters affected model performance in more than one variable, but resulting value ranges differed depending on both, the variable and the performance index (Fig. S2 in the supplement). Performance in Ts and WT was determined by 12 key parameters belonging to seven and six different modules, respectively (Fig. 3). In contrast, snow depth and LAI depended mainly on parameters from their own modules. Large differences in resulting accepted ranges depended on the selected performance index and the considered sub-period: On average, accepted value ranges overlapped with 35%

between different performance indices and between different sub periods of the same variable and with 6% if additionally the differences between different variables were considered (Fig. 4). Radiation and LAI refer to the simplest processes in respect to number of connected parameters (Fig. 3). However, radiation was, together with snow depth, the variable with the strongest average disagreement in parameter value ranges between the different selection criteria (Fig. 4).

In case of eleven parameters, the accepted ranges did not overlap at all (Fig. S2 in the supplement). Four parameters were

sensitive to at least half of the considered variables (Fig. 2): The parameter defining the water retention curve and unsaturated soil hydraulic conductivity ($\psi_a$) affected model performance in variables of all eight considered variables. Moss transpiration





coefficient ($g_{max,moss}$), vascular plant respiration coefficient ($k_{gresp,vasc}$) and litter fall rate ($l_{Lc1}$) were important parameters for not only LAI and NEE, but also H, LE and WT, $g_{max,moss}$ and $k_{gresp,vasc}$, additional for Ts.

The sensitivities of the single parameters are described in more detail in Sect. 3.5. The full table of the correlation coefficients between parameters and performance can be found in the supplement (Table S4).

### 3.2 Confounding and supporting effects of interacting processes

The performances of several variables were connected in supporting and cofounding ways (Fig 5 and 6). Especially ME of LE and WT were strongly connected, but also ME of LAI had an impact on the performance in many other variables. Trade-offs existed not only between the performances of different variables, but also within a variable, depending on chosen performance

index or seasonality.

The magnitude of vascular plant LAI was strongly correlated with magnitude of LE, WT, H and NEE, especially if daytime and night time values were considered (Fig. 5). Thereby the lowest ME in day and night time NEE, as well as ME and dynamics of H, went along with a slight underestimation, and for LE and WT with a slight overestimation of vascular plant LAI. Best performance for WT dynamics was reached if the magnitude of vascular plant LAI was correct (Fig. 6). A noticeable existence

of the vascular plants (LAI ME > −0.4) increased the fit in NEE $R^2$ to at least 0.2, but this was not a necessary precondition for good NEE performance (Fig. 6). Highest performance in dynamics of WT, H and Ts in the upper layer coincided with a good fit in NEE magnitude (Fig. 6). This relationship was even stronger if these variables were compared to ME in NEE night time and NEE daytime.

A correct representation of WT dynamics and depth coincided with high performance in H dynamics and a correct or slightly

underestimated H (Fig. 5 and 6). A small ME in H correlated with high performance in WT dynamics. Performances in soil temperatures of different layers were strongly correlated with each other in both, dynamics and magnitude.

Underestimation of LE was connected to an overestimation of H, but also to better dynamics in H (Fig. 5). ME in Net radiation was positively correlated with ME in H. A good fit between modelled and observed snow depth did not correlate with the performance in any other variable. The only exception was a negative correlation between the dynamics in snow depth and H,

if exclusively performance during spring time was considered (Fig. S3 in the supplement).

Trade-offs existed not only between different variables but also between different performance indices of the same variable. Especially for snow, Rn, and in case of some parameters also for Ts, accepted ranges were contradictory depending on whether $R^2$ or ME was chosen. In case of moss albedo ($a_{pve,moss}$) and aerodynamic resistance dependency on LAI ($r_{alai}$), the ranges also strongly depended on the season during which the variable was considered. For two aerodynamic resistance and one soil

parameter ($z_{0M,snow}$, $c_{H0,canopy}$, $s_k$) ranges differed between $R^2$ of actual values and $R^2$ of accumulated values.

Additional to the uncertainty from unambiguous parameter ranges, further uncertainty results from equifinalities between parameters.



### 3.3 Equifinalities

Parameters were strongly inter-correlated, often with several parameters, and often from different modules. Equifinalities can hinder the identification of sensitivities, which was especially true for the basic selection: Despite reducing the number of runs

by 97.5%, posterior and prior ranges differed hardly (Table S5 in the supplement). Instead certain value triples for photosynthetic efficiency ($\varepsilon_{L,vasc}$) with the respiration coefficient ($k_{gresp,vasc}$) and with the storage fraction for plant regrowth in spring ($m_{retain}$) were crucial for the survival of the vascular plant layer. Certain value pairs for the moss transpiration coefficient ($g_{max,moss}$) with the shape parameter of soil water retention ($\psi_a$) were crucial for a reasonable water table depth.

Equifinalities existed not only between parameters from the same modules, but even more often between parameters from

different modules (Fig. 7). Parameters defining radiation interception, soil temperature, aerodynamic resistance, transpiration, and soil hydrology correlated with exclusively parameter from different modules. Parameters defining radiation interception were mostly correlated with parameters defining aerodynamic resistance. Only in case of plant and SOC decomposition parameters, equifinalities existed mainly between parameters of the same modules.

Except $\rho_{smin}$, all sensitive parameters and further other parameters were detected to correlate with up to five other parameters

in the final selections, $\psi_a$ correlated with even seven others (Fig. 2). Two parameters had very strong correlations ($R^2 \geq 0.3$) with two other parameters each, which belong to different modules ($\psi_a$ with $c_{H0,canopy}$ and $g_{max,moss}$ and $a_{pve,moss}$ with $z_{0M,snow}$ and $r_{alai}$) (Table S6 in the supplement).

### 20 3.4 Usefulness of measurement variables

All available measured variables (NEE, LAI, LE, H, Rn, Ts, WT and snow depth) were helpful in constraining parameter ranges (Fig. 2). None of the supporting effects was strong enough, to make one variable fully replaceable by another. Even for the strongest correlation between soil temperatures of the different layers, the remaining uncertainty in one temperature when knowing the other would be in the magnitude of 0.5°C, which corresponds to more than 25% of the total uncertainty resulting

from the tested parameter ranges (Fig. 5).

13 parameters could be unambiguously constrained to a more narrow range, as their resulting ranges were well overlapping (Fig. S2 in the supplement). The performance on each variable was correlated with many parameters from different processes (Fig. 3). The highest number of correlations was detected for the performance in WT and Ts, which constrained 12 parameters from different modules. Also the available data for LE, H, and NEE constrained many parameters.

30 Still, large uncertainty remained due to equifinalities and differences in accepted ranges: The largest uncertainty was caused by a parameter defining the shape of the water retention curve (air entry, $\psi_a$). As this was the only calibrated parameter of the





water retention curve, it determined the unsaturated hydraulic conductivity of the soil. $\psi_a$ was sensitive to all considered variables and had many strong interactions with other parameters, while it was not possible to constrain it to an unambiguous value range (Fig. S2 in the supplement). Therefore it would be of great value to be able to deduce such parameters from additional measurements. This applies also to following parameters, which could not be constrained unambiguously: Leaf litter

fall rate of vascular plants during the growing season ($l_{Lc1}$) was the second most sensitive parameter, affecting the performance in NEE, H, LE and WT. Moss albedo ($a_{pve,moss}$), aerodynamic resistance dependency on LAI ($r_{alai}$) and transpiration coefficients ($g_{max,vasc}$, $g_{max,moss}$, $g_{maxwin}$) had similar importance, due to their equifinalities to other parameters. Plant respiration ($k_{gresp,vasc}$) had strong sensitivity, but could be constrained unambiguously by the available data.

**3.5  Detailed description of sensitivities and interactions per process**

Detected sensitivities, connections between performances, and equifinalities showed all strong interactions between the different processes and parameters of different modules. Connections existed between all variables and modules, but most strongly interlinked were LE with WT, Rn with H and Ts (Fig. 2). H, LE, WT were also linked to each other and to NEE. The impact of the plant is further reflected in the correlations between performances in LAI with performances in many other

variables (Fig. 5). The implications on the performance for each considered variable will be described in the following sections.

**3.5.1 Water level depth and soil moisture conditions**

Performance in water level depth was determined by 12 key parameters (Table S4 in the supplement). It was most strongly connected to the shape of the soil water retention curve ($\psi_a$) as well as to the transpiration coefficients for mosses and winter transpiration ($g_{max,moss}$, $g_{maxwin}$). The transpiration coefficient from vascular plants played a smaller role due to the high

sensitivities of parameters defining the growth and therefore magnitude of the vascular plant (i.e. $k_{gresp,vasc}$, $m_{retain}$, $l_{Lc1}$). Equifinalities existed between several of these parameters.

$\psi_a$ had strong effect on the performance of all variables and several strong equifinalities, in particular with parameters defining aerodynamic resistance and transpiration; On the other hand $\psi_a$ could not be constrained to an unambiguous range and was therefore the parameter causing the largest overall uncertainty (Fig. 2).

Performance in WT was further sensitive to parameters defining aerodynamic resistance, i.e. $r_{alai}$ and $c_{H0,canopy}$. Both parameters had equifinalities with $\psi_a$ and moss albedo ($a_{pve,moss}$) as well as with timing of snow melt ($m_T$) and thermal conductivity of snow ($s_k$). Also the distance between drainage ($d_p$), showed some sensitivity.

**3.5.2 Transpiration and evaporation**

The nine most important parameters for WT performance were also key parameters for LE ($\psi_a$, $g_{max,vasc}$, $g_{max,moss}$, $g_{maxwin}$,

$k_{gresp,vasc}$, $m_{retain}$, $l_{Lc1}$, $r_{alai}$, $c_{H0,canopy}$). This explains the strong correlation between the performance in WT and LE ME (Fig. 5)





and shows the connections with plant, WT and H. Another parameter, sensitive to LE was the roughness length of snow ($z_{0M,snow}$), belonging to the aerodynamic resistance module and correlating with moss albedo, hinting to the connections between LE and R associated processes.

Dynamics in WT and LE, but also magnitude of H was improved if the transpiration coefficient was on its lower range in case of mosses and on its upper range in case of vascular plants (Fig. S2 in the supplement). Despite the lower values for mosses, transpiration prior criteria selection was dominated by mosses, due to their higher LAI and coverage (Fig. S4 in the supplement).

Crucial for LE performance was also a parameter defining the aerodynamic resistance of the canopy under stabile conditions ($c_{H0,canopy}$): a very small value improved the $R^2$ of LE and spring LE, but downgraded $R^2$ of accumulated LE and of winter radiation.

Spring LE was overestimated in most of the runs (see Fig. S1 in the supplement). The strongest sensitivity on spring LE was by the coefficient for winter transpiration ($g_{maxwin}$): the higher the better $R^2$ and ME. Together with ($z_{0M,snow}$) this was also the most important parameter for winter LE.

### 3.5.3 NEE & LAI

Seven of the nine parameters which were common for LE and WT were also among the most effective parameters for NEE ($\psi_a$, $g_{max,moss}$, $g_{max,vasc}$,, $k_{gresp,vasc}$, $m_{retain}$, $l_{Lc1}$, $r_{alai}$) and belong to four different modules: plant, transpiration, soil hydrology and aerodynamic resistance (Table S4 in the supplement). However the most sensitive parameter for NEE was the rate coefficient for heterotrophic respiration ($k_{l1}$), which was especially important for night time NEE. Further sensitive parameters for night time NEE were the growth respiration coefficient for mosses ($k_{gresp,moss}$) and the temperature dependency coefficient for heterotrophic respiration ($t_{min}$).

The rates of photosynthesis and its temperature dependence ($\varepsilon_{L,vasc}$, $\varepsilon_{Lmoss}$, $p_{mn,vasc}$) were key parameters for LAI, NEE magnitude or temporal NEE dynamics, respectively. Many strong interactions existed between plant parameters, which were especially visible in the basic selection (see Sect. 3.3).

The rate of leaf litter fall during the growing season $l_{Lc1}$ was one of the parameters with the highest concern, due to its sensitivity on many different processes, its equifinalities and as it could not be constrained to an unambiguous solution (Fig. 2). Resulting ranges for $l_{Lc1}$ differed especially between the different performance indices within NEE and within LAI, but also between NEE and LAI (Fig. S2 in the supplement).

### 3.5.4 Sensible heat fluxes, soil temperatures and net radiation

Many inter-connections existed between H, Ts and Rn, but all three were also linked with LE, WT, snow and NEE. A snow parameter, determining the timing of snow melt ($m_T$) was the most crucial parameter for heat fluxes, not only in spring time,





but also for the whole year period. Further, $m_T$ was important for Ts in spring time (cf. Sect. 3.5.5). The shape of the soil water retention curve ($\psi_a$) was the second most sensitive parameter for both variables.

The aerodynamic resistance dependency factor on LAI ($r_{alai}$) was the most sensitive parameter for Ts, and affected also LE, WT and night time NEE, while it strongly correlated with moss albedo ($a_{pve,moss}$), the third most sensitive parameter for H and

most sensitive parameter for Rn. Accepted ranged for $r_{alai}$ contradicted within the soil temperature variables, depending on the chosen performance index and considered season: high values were important for Ts ME and $R^2$ during winter, but low ones improved Ts $R^2$ during spring and during the whole period. Therefore, $r_{alai}$ was the parameter causing the largest overall uncertainty after $\psi_a$. This was followed by $a_{pve,moss}$, which had low values for accepted ranges in case of H, Rn and Ts during the whole period, but high values in case of winter H and Rn. It further showed strong equifinialites with the roughness length

of snow ($z_{0M,snow}$), which was the second most sensitive parameter for Rn, but also affected H and LE. The coefficient for thermal conductivity of snow ($s_k$) affected Rn and Ts, but not H.

The thermal conductance coefficient of soil organic material ($h_2$), the lower boundary mean temperature ($T_{amean}$), the snow melt dependency to radiation coefficient ($m_{Rmin}$) and the density of new and old snow ($\rho_{smin}$, $S_{dw}$) affected only soil temperatures, the latter two also snow depth.

Parameters defining moss and winter transpiration ($g_{max,moss}$, $g_{maxwin}$) and the growth respiration coefficient of vascular plants with its effect on vascular plant biomass and LAI ($k_{gresp,vasc}$) were sensitive to Ts, $g_{max,moss}$ and $k_{gresp,vasc}$ also to H. The most important parameter for LE, $c_{H0,canopy}$ was another key parameter for Rn and H.

### 3.5.5 Snow

The temperature coefficient in the snow melt function ($m_T$) was the most important parameter for ME in snow and determined

timing of snow melt. However, resulting parameter ranges did not overlap between the different performance indices within the snow depth variable and between different other variables. A longer lasting snow cover (low $m_T < 3$) was crucial for spring H and reduced mean error in snow depth, but lowered $R^2$ values in spring Ts and snow depth. $m_T$ interacted with another snow parameter ($T_{RainL}$) as well as with parameters from the temperature and transpiration module ($T_{amean}$, $g_{maxwin}$). The density coefficients for old ($S_{dw}$) and new snow ($\rho_{smin}$) had medium effect on snow depth performance, and affected also spring and

winter soil temperatures in all layers, but the latter could be unambiguously constrained by the available data.

### 4 Discussion

Unlike many previous sensitivity studies for $CO_2$ modelling that often focus on only one or few calibration variables and parameters of the associated module, we considered many different abiotic and biotic measurements (NEE, LAI, Rn, Ts, H,

LE, WT and snow depth) to investigate the interactions between various processes (SOC decomposition, plant growth related





processes, radiation interception, soil temperature, aerodynamic resistance, transpiration, soil hydrology and snow) in a peatland ecosystem.

Similarly to results from a forest modelling study (Tian et al., 2014) and a N2O study using CoupModel on a drained peatland forest (He et al., 2016), we found that processes were sensitive to parameters from several different modules. Together with

the discovered supporting effects between model performances in different variables, this confirms the connections and dependencies between different processes as implemented in the model (cf. Model description and equations, Sect. 2.3, Table 2 in the supplement and Janson and Karlberg, 2010). The many interactions between parameters of both, between the same and also between different modules, reveal the dependency of constrained parameter ranges as well as parameter sensitivities to model structure, calibration setup and parameters with fixed values. This implies a limited transferability of parameter values

between models and even between studies using the same model in a different configuration. Yet, it is quite common practice to adopt at least some parameter values from other modelling studies (e.g. Frolking et al., 2002, Yurova et al., 2007; St. Hilaire et al., 2010; Wania et al., 2010; Gong et al., 2013; Kim et al., 2014; Kurnianto et al., 2014; Zhu et al., 2014), which includes the usage of model default values that were estimated under a different model configuration.

Further, the strong interactions across different modules emphasize the importance of measurements of ancillary data

additionally to the variable of interest and model input data (meteorological and SOC data). Measurements of NEE, LAI, LE, H, Rn, Ts, WT and snow were all found to be valuable for constraining parameters from several different modules and can therefore reduce uncertainty in model predictions. Further constraint would be possible, if especially additional water content or soil hydraulic properties were measured.

Beside parameter uncertainty, also uncertainty in model structure and in measured input and calibration data contribute to

model uncertainty (Thorsen et al., 2001; Beven and Freer, 2001). This was tested for other peatland models (e.g. model structure: Tang et al., 2015; input drivers: Wania et al., 2009; St-Hilaire et al., 2010; Grant et al., 2011, Kim et al., 2014), but goes beyond the scope of this study.

## 4.1 Parameter sensitivity

The gained knowledge on parameter sensitivities can help to simplify future calibrations (Saltelli et al., 2000), by focussing

on the most striking parameters and narrowing the ranges for parameter which could be successfully constrained. Further it helped to identify process interactions. Especially abiotic processes were strongly inter-linked, but also biotic variables showed sensitivities to parameters from up to seven different modules, suggesting that parameter sensitivities and model performance of a certain process depend on which other modules are considered in the model and in the calibration. This is an important finding, as many studies investigate the sensitivity of often only few parameters from mainly the same module as the output

variable (e.g. Yu et al., 2001; Frolking et al., 2002; Belassen et al., 2010; Wania et al., 2010; Morris et al., 2012; Wu and Blodau, 2013; Zaho et al., 2013; Zhu et al., 2014). The knowledge on these dependencies can help modellers to select an



appropriate model including the parameters, processes and modules which need to be considered together, depending on the variable of interest.

While the existence of interactions between the processes and their parameters is supposed to be less dependent on site conditions and model structure, the exact shape of the connections as well as constraint parameter ranges might strongly depend
on these factors.

One or more of the following parameters that we identified as  most influential, correspond to key parameters in other studies using other models and partly different ecosystems: The respiration rate coefficients, radiation use efficiency, transpiration coefficients or the soil water retention capacity were among the most sensitive parameters for NEE, its components, or yield, respectively, in e.g. the PCARS (Frolking et al., 2002) and the GUESS-ROMUL (Yurova et al., 2007) model on peatland, the
SiB v2.5 model on a forest area including some wetlands (Prihodko et al., 2008), the LPJ-GUESS model on forest and herbaceous vegetation (Pappas et al., 2013), the EPIC model on cropland (Wang et al., 2005), the BIOME-BGC model for different tree species (Tatarinov and Cienciala, 2006), or the ACASA (Staudt et al., 2010), the 3-PG (Esprey et al., 2004; Xenakis et al., 2008), the FORUG (Verbeeck et al., 2006) or the DRAINMOD-FOREST (Tian et al., 2014) model on forest. These sensitivities seem to be therefore quite independent of model structure, included processes and parameters used for
calibration. The resulting value ranges of these parameters should be compared between ecosystems and models to find out to what extent they can be related to site conditions and therefore used for predictions and upscaling.

However, one has to bear in mind that resulting constrained ranges might be connected to the environmental scenario (Hidy et al., 2012; Ben Thouhami et al., 2013; Sulman et al., 2013) and the chosen prior distributions of the parameters (e.g. Tatarinov & Cienciala, 2006). Further, our results have shown that the parameter ranges depend on model structure, on the selection of
parameters for calibration and on the selected acceptance criteria. Thereby, not only the selected variable, but also the selected sub-period was relevant, as has been shown by other studies as well (e.g. Prihodko et al., 2008; Van Huisteden et al., 2009; Safta et al., 2014).

## 4.2  Confounding and supporting effects of interacting processes

Criteria selection is a subjective choice of the modeller if multiple output variables are available. The identified supporting
effects and trade-offs between the performances in different variables allow modellers to assess the implications of a certain criteria on model performance and parameter ranges and to choose criteria according to the processes of interest.

Usually, LE is assumed to be closely connected to NEE due to the coupling of transpiration and carbon assimilation in vascular plants (e.g. Schulze et al., 2006), but has also been shown to correlate for mosses (e.g.  Robroek et al., 2009). Our study reveals much stronger relations between parameters defining H and NEE, than between LE and NEE. Trade-offs between performance
in LE and NEE were also found by Staudt et al. (2010) and Prihodko et al. (2008) in a forest and a forest complex including wetlands. However, only the effect of parameters, not the effect of input variables on these processes were tested in both studies, as well as in ours. Such a confounding effect might also be the effect of a process not implemented in the model, like



e.g. evaporation from open water bodies formed during spring and early summer, which might also explain the model data mismatch in LE during mid-April to mid-June (Fig. S1 in the supplement).

Trade-offs existed not only between different variables but also within the same variable, depending on whether ME, $R^2$ of actual or $R^2$ of accumulated values was chosen and which season was considered. This implies that the problems of a subjective
criteria selection also exist if only one time series variable is considered.

Also several supporting effects were detected, indicating that some measurement variables can partly compensate absence or low resolution of a connected variable, even though they were not strong enough to make one variable fully redundant. For example, LAI measurements could reduce uncertainty in model predictions of the magnitudes of NEE, LE, H and WT on locations where these variables are not available. Tight relationships between plant and LAI, soil hydrology, C-fluxes and soil
temperatures have been found by other model sensitivity studies as well (e.g. Ben Thouhami et al., 2013; Quillet 2013; Tian et al., 2014; Sándor et al., 2016) and strong correlations between LAI and NEE (Lund et al., 2010), and NEE and water availability (Reichstein et al., 2007) have also been found by data syntheses of eddy covariance sites. These relationships can be explained by the many dependencies between LAI and e.g. photosynthesis, transpiration, heat insulation and water uptake (Schulze 2006), of which several are also implemented in the model (see model description and equations, Sect. 2.3, Table S2
in the supplement and Jansson and Karlberg, 2010).

Other examples for detected supporting effects indicate that if H fluxes are available, the model is constrainable to produce improved WT dynamics, even if WT measurements were missing. High resolution of soil temperature measurements in one layer are sufficient to model good temperatures if just the magnitude of soil temperature in an upper and a lower layer is known, e.g. due to short time or low resolution measurements.
The knowledge on supporting effects helps modellers in their site selection and in uncertainty estimation of model predictions depending on available ancillary data. It further can help experimentalists in their decisions which variables should and which need to be measured if the site should be usable for model constraint.

### 4.3 Equifinalities

The fit of model output to measured data in complex models is often not driven by a particular parameter but instead by
interactions among parameters (e.g. Beven and Freer, 2001, which was also the case for several parameters in our study, hindering the constraint of parameters to a more narrow range. Also other carbon modelling studies found, that parameter values and sensitivities depend on the values of other parameters (e.g. Tatarinov & Cienciala 2006; Verbeeck et al., 2006; Quillet et al., 2013). This implies that especially if only few parameters and processes are calibrated (as in e.g. Yu et al., 2001; Wania et al., 2010, Zhu et al., 2014; Kim et al., 2014, Tang et al., 2015), resulting constrained ranges might not be comparable
and transferable between models differing in their constant parameter values. Many equifinalities were identified, not only between parameters from the same module, but also from different modules. This means that the problem of limited





transferability also applies, if parameters from only one module are calibrated (as e.g. in Wang et al., 2005, Belassen et al., 2010, Wania et al., 2010, Sándor et al., 2016), or if models differ in the structures and implementations of their modules. The knowledge on equifinalities is needed for a better parameter constraint in future calibrations as it allows calibration of the connected parameters in dependency of each other. Another way to respond to identified equifinalities is to calibrate only one

of the connected parameters. However the resulting range will then not be transferable to other models using different values for connected, constant parameters.

Some equifinalities included several parameters, making their visualisation impossible and simple regression an insufficient tool for fully detecting and describing them (cf. Saltelli et al., 2008). These equifinalities need to be further investigated in additional calibrations which incorporate those parameter interactions and constrained ranges which were unambiguous, to

achieve a higher number of acceptable runs. This is needed, because the numbers of accepted runs in the final selections (50) did not allow a much more detailed analysis in such a complex model, as was apparent in comparison with the basic selection: An $R^2$ threshold value of 0.1 was sufficient to identify equifinalities in the basic selection of 1286 accepted runs, but with just 50 accepted runs in the final selections, this threshold value could easily be exceeded by a random distribution, even that a higher threshold value of 0.15 was used. A threshold of 0.15 was on the other hand already too high, to detect for example the

strong relationships between the plant parameters which were only clearly visible in the basic selection. Nevertheless the six equifinalities with $R^2$ of higher than 0.30 are unambiguous and those with lower values are still very useful to design future calibrations to further investigate and describe these equifinalities.

## 4.4 Usefulness of measurement variables

Models can be improved and their uncertainty reduced by calibrating their parameters to measurement data (e.g. Friend et al.,

2007; Wang et al., 2009; Williams et al., 2009). We tested the usefulness of several measurement variables (NEE, LAI, LE, H, Rn, Ts, WT and snow depth) and found all contributing to a better parameter constraint. Thereby none of the variables could be fully replaced by another. Due to the strong interactions and as parameters of each module were constrained by several different variables, ancillary variables are valuable even if only one certain process is of interest. In case of snow, our results suggest that data on snow cover might be sufficient, if snow depth is not available.

In a forest site simulation with the ORCHIDEE model, H and Rn were found to be redundant for constraining energy balance parameters if NEE and LE were available (Santaren et al., 2007). In contrast, some energy balance related parameters in our study were constrained by exclusively Rn and H, or additionally by LE but with different resulting ranges. This reveals the usefulness of Rn and H measurements for model constraints and shows that variables which might have been identified as redundant in one study could be of high importance on another ecosystem or for another model calibrating a different parameter

selection.

Several influential parameters could not be unambiguously constrained or showed equifinalities and need additional measurements to be further investigated. This includes soil water content or soil water retention properties, as well as canopy





albedo and leaf litter fall during the growing season. Except for water retention properties these variables are needed as time series throughout the year. A more detailed discussion of the benefit of such measurements can be found in the following sections.

## 4.5  Detailed discussion of sensitivities and interactions per process

The parameters that were identified as most influential or that showed the strongest equifinalities were related to soil hydrology and water content, to a stable representation of the plant, to radiation, temperature and heat fluxes or to snow.

The introduced index to measure parameter concern includes subjective choices like weighting factors, the choice of considered calibration variables and their sub periods as well as the chosen performance indices. However several tested variations in especially the weighting did not noticeable change the results: $\psi_a$ was always the most important parameter, followed by the group of parameters with medium importance which differed slightly in their ranking among each other.

### 4.5.1 Unsaturated water distribution & soil moisture conditions

Our results suggest that model uncertainty could be greatly reduced if data for either soil hydraulic properties, water content or plant transpiration characteristics were available: Despite available data of detailed WT and LE in our study, large uncertainty remained in simulated water content due to the combined uncertainty in estimates of soil hydraulic properties ($\psi_a$) and plant water uptake ($g_{max,vasc}$, $g_{max,moss}$, $g_{maxwin}$). Their sensitivity to many variables and the high number of equifinalities hindered the constraint of other parameters and therefore the uncertainty reduction in all involved processes. For example this might explain why the water response functions for neither plant assimilation nor soil respiration could be constrained.

The shape parameter of the water retention curve ($\psi_a$) was among the top two most sensitive parameters for NEE, WT, LE, H, Ts, and the third and fifth most sensitive parameter in case of Rn and snow. That confirms the importance of the implemented interactions of soil moisture with water and heat fluxes, soil temperature, assimilation and respiration processes, as reported from empirical studies (Kim and Verma, 1996; Bridgham et al., 1999; Tezara et al., 1999; Kellner, 2001; Flangan and Johnson 2005; Lafleur et al., 2005; Schulze, 2006; Belyea 2009).

Also, the transpiration coefficients ($g_{max,vasc}$, $g_{max,moss}$, $g_{maxwin}$) were among the top 10 most important and influential parameters. In case of vascular plants, they correspond to the stomatal conductance parameter in other models, which was shown to be crucial for modelling NEE, biomass, LE or H in other studies (Esprey et al., 2004 for forest stand volume, Tatarinov and Cienciala, 2006 for NEE and carbon pools; Staudt et al., 2010 for NEE, LE and H; Hidy et al., 2012 for carbon fluxes and LE; Bonan et al., 2011 and Tian et al., 2014 for LH and H). The control of stomatal conductance on transpiration and photosynthesis has also been emphazised by several empiric studies (e.g. Jarvis & Morison 1981, Quick et al., 1992, Tezara et al., 1999, Yordanov et al., 2000).




The strong sensitivity of $\psi_a$, $g_{max,vasc}$, $g_{max,moss}$, $g_{maxwin}$ for many processes is especially remarkable as parameters and parameter combinations could only vary to such an extent that the water level fitted the measurements as restricted by the basic selection. The importance of water table on NEE fluxes has widely been mentioned (e.g. Silvola et al., 1996; Yurova et al., 2007, Kurbatova et al., 2009; Dušek et al., 2012) but our results point out that the knowledge on WT alone is not sufficient for model

calibration and reliable predictions. In addition also measurements of soil hydraulic properties are crucial for model calibration. The usefulness of water retention properties for modelling carbon dynamics was also found by other sensitivity analyses (e.g. Wang et al., 2005; Pappas et al., 2013, Quillet et al., 2013). Nevertheless, many of the available peatland sites in current databases (e.g. European Fluxnet Database Cluster, http://gaia.agraria.unitus.it) still do not contain information on water retention properties or water content.

We therefore strongly recommend experimentalists to include water retention measurements in their experimental set up. Thereby, the horizontal and vertical variablity in peat hydraulic properties needs to be accounted for (Baird et al., 2012, Waddington et al., 2015). Such measurements might also help to resolve the strong equifinalities of $\psi_a$ with transpiration coefficients and a parameter in the calculation of aerodynamic resistance of the plant canopy, defining the minimum exchange under stabile conditions ($c_{H0,canopy}$).

### 4.5.2 C balance of vascular plants

A stable vascular plant that establishes a reasonable amount of biomass every year throughout the simulation period, could only be achieved by certain value combinations for the photosynthetic efficiency ($\varepsilon_{L,vasc}$), the respiration coefficient ($k_{gresp,vasc}$) and the storage fraction for plant regrowth in spring ($m_{retain}$). Despite their high impact in the basic selection, neither

equifinalities, nor sensitivities of these parameters reached high measures in final selections, probably because several parameters were interacting simultaneously. This indicates the need for either calibrating these parameters in dependency of each other or setting at least one of them to a constant value, as the available data was not sufficient to resolve these equifinalities. Many studies on other ecosystems have found NEE or biomass to be strongly sensitive to a parameter corresponding to photosynthetic efficiency ($\varepsilon_{L,vasc}$) (Esprey et al., 2004, Verbeeck et al., 2006; Prihodko et al., 2008; Staudt et

al., 2010; Bonan et al., 2011; Pappas et al., 2013, Tian et al., 2014, Xenakis et al., 2008), but were performed without a simultaneous calibration of parameters related to plant respiration and storage for regrowth. Pappas et al. (2013) discussed a possible overestimation of model sensitivity to photosynthetic efficiency due to processes that are not implemented like the active simulation of plant growth including growth limitations. A strong negative correlation between two of the parameters (plant respiration and photosynthetic efficiency) was also found in a sensitivity analysis using the LPJ model (Zaehle et al.,

30   2005).

Despite their effect on model performance, $\varepsilon_{L,vasc}$, $k_{gresp,vasc}$ and $m_{retain}$ had a low rank in parameter concern, as ranges for these parameters could be narrowed unambiguously due to well overlapping ranges between the different variables. Nevertheless, these parameters would be of high importance for predictions, if none of the constraining variables are available.





Compared to a previous application of the CoupModel on five different open peatlands including different management intensities (Metzger et al., 2015), vascular plants had to have a much more effective C household to produce the measured leaf area given a limited amount of assimilates. This can be realised by low respiration and litter fall losses and a large storage pool for regrowth in spring. Even if respiration losses from vascular plants were 1/10 of the ones used at the sites in Metzger et al.

(2015), the model tended to either underestimate vascular plant LAI, or overestimate $CO_2$ uptake (Fig 2). A possible explanation for the differences in parameter value combination of vascular plants might lie in the vegetation communities. Despite Metzger et al. (2015) included several different types of treeless peatland vegetation communities, none of these sites had a similar vegetation community typical for nutrient poor habitats, consisting of mainly mosses and *Eriophorum vaginatum*, as at Degerö Stormyr. *Eriophorum vaginatum* is known to be much more effective in maintaining C compared to other sedges

and having a highly efficient remobilization from senescing leaves (Shaver and Laundre, 1997; Jonasson and Chapin III, 1985). Uncertainties in measurements and the distribution of modelled respiration over the hours of the day might accelerate or diminish this effect. Explanations by differences in model structure can be excluded, as the same effect was observed when using exactly the same structure (unpublished data). To identify the difference between the sites, which causes the deviations in the combined parameter value ranges, the model need to be applied to further open peatland sites differing in vegetation

community, nutrient status and plant productivity. This might allow finding trends in parameter ranges, which is a necessary precondition for estimation and reducing model uncertainty in predictions on other peatland sites.

Another plant parameter which was important for a stable vascular plant layer and was ranked as one of the overall most important parameters was the rate coefficient for the leaf litter fall during the growing season ($l_{Lc1}$). Probably due to the high number of correlations with other parameters, these correlations did not exceed the threshold value. $l_{Lc1}$ is directly connected

to the filling of the storage pool, but also for maintaining C in the leaves. The strong sensitivity of LAI to $l_{Lc1}$ affects transpiration and thereby water uptake which explains the strong sensitivity to WT depths below −0.2 m and the equifinalities with a transpiration parameter and a parameter describing the response of heterotrophic respiration to water. In Metzger et al. (2015), a value of $l_{Lc1} = 0.01$ day$^{-1}$ could be used site independent. This contradicts the much lower ranges of $l_{Lc1}$ in our study, necessary for acceptable performance in several variables, in particular $R^2$ of LAI, WT depths below −0.2 m and ME of spring

time NEE. However, species in nutrient-poor habitats are associated with longer-lived leaves than those of nutrient-rich habitats (Ryser 1996) and fast growing species (Reich et al., 1992), whereas *Eriophorum vaginatum* in particular is known for long-lived leaves and therefore have a very low litter fall rate (Jonassson & Chapin 1985). Less complex models as the GUESS-ROMUL model which was also applied to this site, use annual accumulated NEE as estimate for litter fall (Yurova et al., 2007) which is therefore directly dependent on site productivity. Only one site in Metzger et al. (2015) had lower annual NEE

compared to Degerö Stormyr, but this is probably a result of the shorter vegetation period at that site, whereas a site with similar annual NEE was formerly drained, so that the soil respiration contribution to NEE is much larger, compensating the larger productivity. A high sensitivity of litter fall rate to plant biomass and soil carbon pools was also found by Xenakis et al. (2008) using the 3-PG model on forest.



### 4.5.3 Sensible heat fluxes, soil temperatures and net radiation

The large number of strong connections between H, Ts and Rn and the equifinalities between their determining parameters indicate the importance to consider, model and calibrate the related processes together. However the constraint of two of the most important parameters (aerodynamic resistance dependency on LAI, $r_{alai}$ and moss albedo, $a_{pve,moss}$), failed not due to different ranges between variables but due to the differences depending on which performance index and season was considered. This emphasises the importance of the subjective criteria choice, even if only one variable is considered.

Accepted values for $r_{alai}$ were exceptionally high (200 s m$^{-1}$ for Ts R$^2$ and 550 to 800 s m$^{-1}$ for Ts$_1$ ME, whereas a $r_{alai}$ of 200 multiplied with the moss LAI of 1.8 leads to an aerodynamic resistance of 360 s m$^{-1}$). Mosses might form a well insulating layer, but still the values are much higher than the aerodynamic resistance estimates for this site (approximately 50 s m$^{-1}$, Peichl et al., 2013) or of a bog in South-Sweden (60 s m$^{-1}$, Kellner, 2001). Price (1991) reported very high resistance, when moss surface moisture is low, e.g. during dry periods, but these values were still lower than ours. A possible explanation might

be an interaction with a non-calibrated, fixed parameter. A high aerodynamic resistance causes better temperature insulation leading to higher summer soil temperatures with lower diurnal oscillations. Further, it leads to strongly reduced soil evaporation and therefore reduced LE, even though this is partly compensated by slightly higher transpiration from mainly mosses, which profit from the higher water contents in upper soil layers. This explains the sensitivities to WT and LE which also supported a higher $r_{alai}$ value. The main cause for the much lower optimum range for dynamics in Ts compared to

magnitude in Ts is probably an overestimation of the diurnal amplitude. A lower moss LAI can reduce this overestimation, but the corresponding parameter was not calibrated to avoid further equifinalities: $r_{alai}$ showed already strong interactions with $a_{pve,moss}$ and $z_{0M,snow}$. The correlations of the conductivity of organic material ($h_2$) with plant, LE and WT parameters might be explained by the dependency of thermal conductivity from peat wetness (Kellner, 2001).

Seasonal differences in moss albedo ($a_{pve,moss}$) could be expected as their radiation reflection properties vary with moss water

content (Graham et al., 2006). However higher values would be expected in summer, when the moss surface is dry and lighter, but our calibration resulted in higher values during spring and winter. These values were much higher (>22%) compared to literature values (11–16.5%, Berglund and Mace, 1972; 16.4%, Zhao et al., 1997; 11%, Kellner, 2001) and therefore rather compensate for values of interacting parameters (in particular $z_{0M,snow}$ and $r_{alai}$) or not implemented processes. Especially the effect on winter H and Rn might result from the strong interaction with $z_{0M,snow}$, as the mosses in winter are covered with a

thick snow cover, so that their albedo shouldn't show any sensitivity in winter. Further, H in spring tended to be overestimated, which would be compensated by a high albedo during this time, but might be caused in the real world by open water over frozen soil, which was not realized in the model. Interestingly, albedo of vascular plants did not show any sensitivities, neither during vegetative stage ($a_{pve,vasc}$), nor after start of senescence ($a_{pgrain}$) when a higher value would have been expected due to





leaf yellowing. Direct measurements of plant albedo were not available in this study. A time series observation of those would be very helpful for clarification, as this parameter is known to vary substantially within and between peatlands (Belyea et al., 2009).

### 4.5.4 Snow

The model performance in simulating snow depth was not connected to performance in any other variable, except to performance in H if exclusively spring time values were considered. This was surprising, as the uncertainty for timing of snow melt ranged for about two weeks but determined the start of temperature rise, water table dropping and biotic activity. A possible explanation might be the poor ability of snow depth $R^2$ and ME to assert a good fit in duration of snow cover. This is supported by the fact that the most important parameter for timing of snow melt ($m_T$) strongly affected performance in dynamics of H, NEE and Ts during spring time. Parameters defining timing of snow depth might be better constrained if future calibrations include an additional variable with a stronger conclusiveness to the timing of snow melt, e.g. by a boolean time series indicating if snow cover is present or not. It needs to be tested if this could also help to solve the disagreements in value ranges between the performance indices in case of the density coefficient of old snow ($S_{dw}$) which caused in combination with $m_T$ the low average overlap within snow depth sensitive parameters.

According to Jansson and Karlberg (2010), a high value for $m_T$ (4–6 kg °C$^{-1}$ m$^{-2}$ day$^{-1}$) could be expected for open fields. A possible explanation for the low accepted values (<3 kg °C$^{-1}$ m$^{-2}$ day$^{-1}$) of $m_T$ in case of criteria on H in contrast to the high values if criteria were on Ts, could be that high values compensate for overestimated spring time H (cf. Fig. S1 in the supplement). However, the overestimation of spring H might be connected to different reflection properties of mosses during spring time or to missing consideration of radiation reflection and evaporation from open water which might be formed during snow melt on still frozen soils. The latter is further supported by the underestimation of LE during April and May (Fig. S1 in the supplement), which cannot be connected to underestimated plant transpiration, as the model even tended to overestimate $CO_2$ uptake during this period.

## 5 Conclusions

$CO_2$ models are often calibrated on NEE as only measurement variable. We investigated the interactions between different abiotic and biotic processes and their parameters, as well as the implications and usefulness of data on not only NEE, but also LAI, sensible and latent heat fluxes, radiation, water table depth, soil temperatures and snow depth for model calibration on a boreal peatland. Different processes and their parameters as well as model performance between different observation variables were strongly interlinked. This needs to be taken into account in model calibrations and when transferring calibrations results between models differing in their structure or in their constant parameters.





The key parameters identified will help to simplify future model calibrations by selecting only the most influential parameters for the variable of interest and using a narrower range for the constrained parameters. This means a simpler calibration and faster computation and in turn, allows the inclusion of a more detailed investigation of a process of certain interest. On the other hand, our results revealed the strong dependence of constrained parameter ranges to other parameters and to the chosen

criteria. This means, that a study aiming to understand and interpret parameter values need to calibrate processes and parameters of many different modules, using a wide range and multiple criteria on various observation variables.

Parameter interactions were found to be more important than parameter value ranges, revealing the need for accounting for equifinalities: Either by calibrating correlated parameters in dependence of each other or by calibrating only one of the correlated parameters. The latter will lead to a narrower constrained range, but this range might not be transferable to other

sites and other models.

The gained knowledge on trade-offs will be useful to avoid modelling studies with too many purposes and helps model users assessing the implications of their criteria choice. The validity of calibrated models is always restricted and robustness of obtained parameters should be questioned.

The identified supporting effects between some variables indicated that some measurement variables can partly compensate

absence or low resolution of the connected variable. This information tells experimentalists which measurement variables are helpful and which are obligatory if a certain process should be understood from the underlying regulating principles. It further helps modellers to decide if a site has enough available data for model calibration and to estimate uncertainties in model predictions depending on available ancillary data.

All observed calibration variables (NEE, LAI, sensible and latent heat fluxes, net radiation, soil temperatures, water table depth

and snow depth) helped for model constraint and interpretation. They should therefore be measured on sites used for calibration of complex process oriented models. Additional measurements of, in particular, soil hydraulic properties or water content would largely reduce uncertainty and help for a better parameter constraint.

**Code and data availability**

The model and extensive documentation can be downloaded from the CoupModel homepage http://www.coupmodel.com/.

The source code can be requested for non-commercial purposes from Per-Erik Janson (pej@kth.se). The simulation files including the model and calibration setup, the used parametrisation and corresponding input and validation files can be requested from Christine Metzger (cmetzger@kth.se). They cannot be made freely public available, as they include climate and site data that require authorisation from the data owners.

The flux data and ancillary data are available from the European Flux Database Cluster (http://www.europe-fluxdata.eu/), site

name: Degerö, Site code: SE-Deg, with open data access for the years 2001–2006, and restricted data access (the Principal Investigator of the site has to authorize the data request) for the years 2007–2015.



## Acknowledgements

This study was financed by FORMAS (grant no. 2009-872), project LAGGE – Landscape Greenhouse Gas Exchange – Integration of Terrestrial and Freshwater sources and sinks, coordinated by Leif Klemedtsson.

The authors thank Mikaell Ottosson Löfvenius and Jörgen Sagerfors from the Swedish University of Agricultural Sciences for

5   their great effort in collecting and providing the model input and calibration data. Climate data from 1990 to 2000 was provided by "reference climate monitoring program at Vindeln experimental forests, SLU, Umea".

The measurements were funded by the Swedish Research Council for Environment, Agricultural Sciences and Spatial Planning (grant no. 21.4/2003-0876 and no. 2007-666) and the Swedish Research Council (grant no. 621-2003-2730). We also acknowledge the Kempe Foundation for the grants supporting the micrometeorological instrumentation. Support from the

10   ICOS Sweden (Integrated Carbon Observation System) and SITES (Swedish Infrastructure for Ecosystem Research) research infrastructure (Swedish Research Council) is also acknowledged.



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



**Tables**

**Table 1. Measurement data used as model input**

| Variable | Period | Resolution as used for model input[a] | Method[b] | Measurement height |
|---|---|---|---|---|
| Global radiation | 1991–2013 | Hourly; 1991-2000: hourly values calculated from daily values by assuming a sinusoidal distribution between 07:30 and 19:30. | 2001-2013: Li200sz sensor (LI-COR, Lincoln, NE, USA) | 3m |
| Air temperature | 1991–2013 | Hourly | MP100 temperature and moisture sensor (Rotronic AG, Bassersdorf, Switzerland) equipped with a ventilated radiation shield | 3 m |
| Relative humidity | | Hourly; 1991-2000: hourly values calculated from daily values by assuming equally distribution during each day | MP100 temperature and moisture sensor (Rotronic AG, Bassersdorf, Switzerland) equipped with a ventilated radiation shield | 3 m |
| Precipitation | 1991–2013 | Hourly; 1991-2000 and November to April: the total daily precipitation was assumed to fell at 12:00 each day | Rainfall tipping-bucket (ARG 100, Campbell Scientific, Logan, UT, USA). | 1 m |
| Wind speed | 1991–2013 | Hourly; 1991-2000: hourly values calculated from daily values by assuming equally distribution during each day | 2001-2013:3-d wind anemometer (Gill Instruments Ltd., Hampshire, UK) | 1.8 m |
| C content per soil layer | 1994 | One time in 1994 | Every 4 cm between 0 and –32 cm, and every 12 cm between –60 and –338 cm | 0 to –338 cm |

[a]: Measurement resolution was the same or higher, except where mentioned differently.

[b]: The method description of meteorological input data applies to the climate station at the site. For gap-filling and for the pre-evaluation period, the data was obtained from the nearby standard climate station (Svartberget field station).



**Table 2: Measurement data used for model calibration**

| Variable | Period | Resolution as used for calibration | Method | Measurement height |
|---|---|---|---|---|
| NEE | 2001–2012 | hourly | EC system, consisting of a three-dimensional sonic anemometer (1012R3 Solent, Gill Instruments, UK; heated during winter months) and a closed path infrared gas analyzer (IRGA 6262, LI-COR, Lincoln, Nebraska USA). Fluxes were calculated by the EcoFlux software (In Situ Flux AB, Ockelbo, Sweden) according to the EUROFLUX methodology (Aubinet et al., 1999, Sagerfors et al., 2008, Nilsson et al., 2008) | 1.8 m |
| LE & H | 2001–2009 | hourly | Same EC system as above (Peichl et al., 2014) | 1.8 m |
| Water table | 2001–2009 | daily | Float and counterweight system attached to a potentiometer (Roulet et al., 1991) | |
| Soil temperature | 2001–2012 | hourly | TO3R thermistors mounted in sealed, waterproof, stainless steel tubes (TOJO Skogsteknik, Djäkneboda, Sweden) in a lawn community 100 m northeast of the flux tower | −2 cm, −42 cm |
| Net radiation | 2001–2011 | | NR-Lite sensor (Kipp&Zonen, Delft, the Netherlands) | 4 m |
| Snow depth | 2001–2012 | daily | Sr-50 ultrasonic sensor (Campbell Scientific, Logan, UT, USA) nearby the flux-tower | |
| LAI of vascular plants | May–Sept. 2012 | biweekly | Destructive sampling (Peichl et al., 2015) | |



**Table 3: Different criteria sets for the selections of accepted runs**

| Main component | Variable | $R^2$ | Mean error (ME) |
|---|---|---|---|
| Basic selection (these criteria are applied additionally in all following criteria sets) | WT < −0.2 m | ≥0.40 | ±0.02 m |
| | LAI vascular plants | ≥0.40 | ±0.02 $m^2\,m^{-2}$ |
| | Daytime NEE | | ±2 $gCO_2$-C $m^{-2}\,d^{-1}$ |
| NEE | Accumulated NEE | ≥0.98 | |
| | Daytime NEE | | ±0.02 $gCO_2$-C $m^{-2}\,d^{-1}$ |
| | Night time NEE | | ±0.07 $gCO_2$-C $m^{-2}\,d^{-1}$ |
| Sensible heat | H | | ±3·$10^5$ J $m^{-2}\,d^{-1}$ |
| | Accumulated H | ≥0.97 | |
| Latent heat | LE | | ±1·$10^5$ J $m^{-2}\,d^{-1}$ |
| | Accumulated LE | ≥0.98 | |
| Net radiation | Net radiation | ≥0.82 | ±4·$10^4$ J $m^{-2}\,d^{-1}$ |
| Soil temperature | Temperature −2 cm | ≥0.95 | ±0.22 °C |
| | Temperature −42 cm | | ±0.22 °C |
| Snow | Snow depth | ≥0.76 | |
| Water table | WT < −0.15 m | ≥0.51 | |

**Figures**

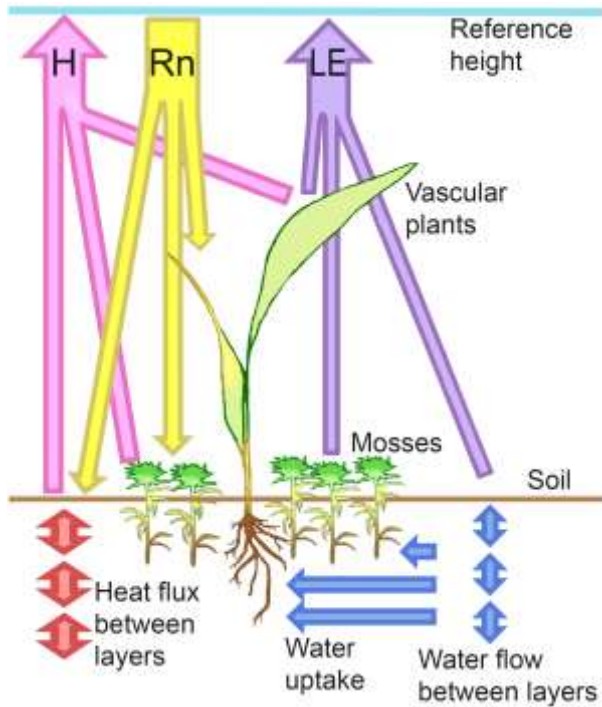

Figure 1. Energy flux partitioning and related soil water flows in the CoupModel as applied to a peatland using two plant canopies and root systems. Rn: Incoming radiation, LE: latent heat fluxes (sum of actual transpiration, interception evaporation and soil evaporation), H: sensible heat fluxes





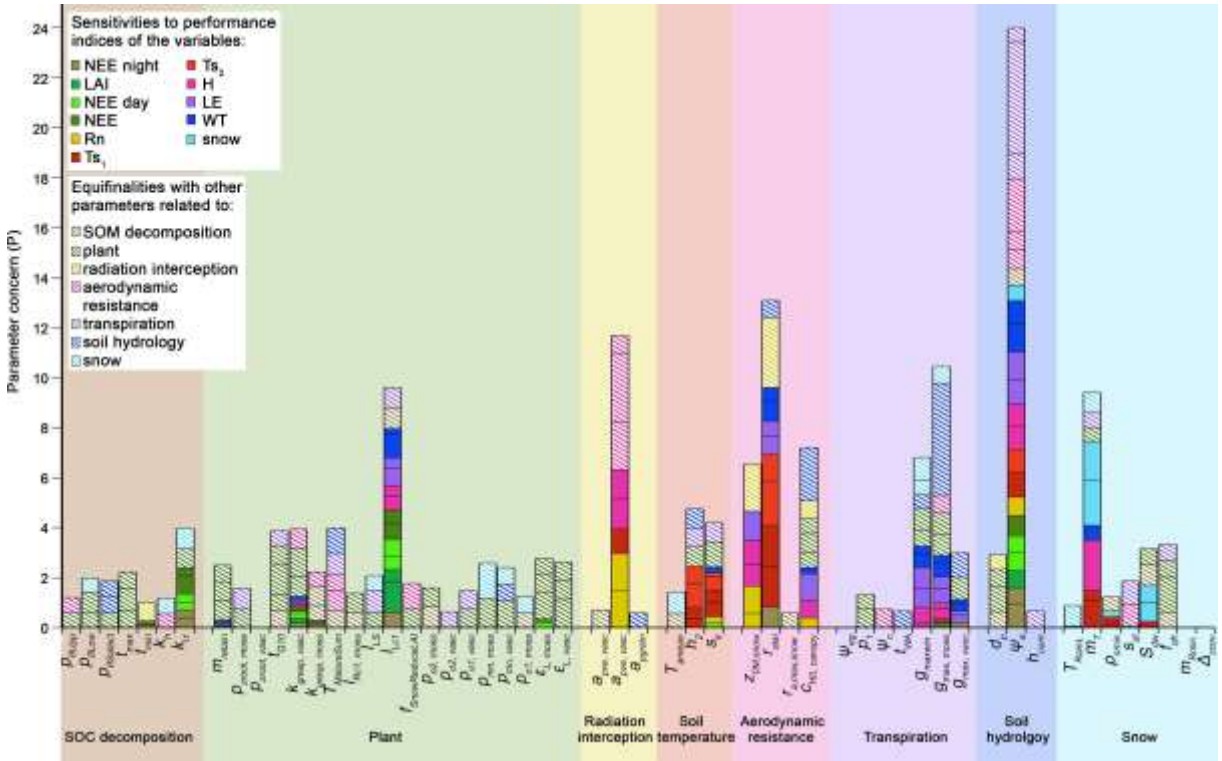

**Figure 2. Parameter concern is shown on the y axis as sum of equifinalities (hatched) and sensitivities that could not be constrained unambiguously (solid). The x-axis shows the parameters, which belong to the module of the background colour.**



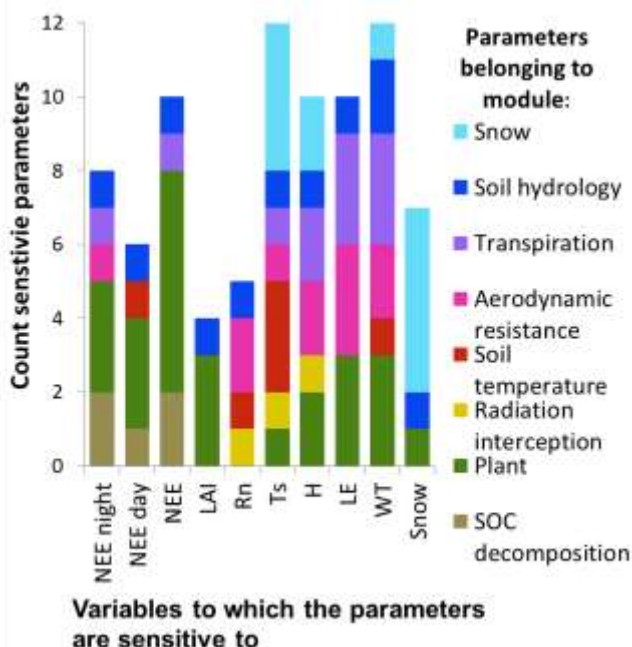

**Figure 3. Connections between processes and parameters of different modules. The y-axis shows the count of parameters from the different modules (colours) that are sensitive to model performance in the various variables (x-axis).**



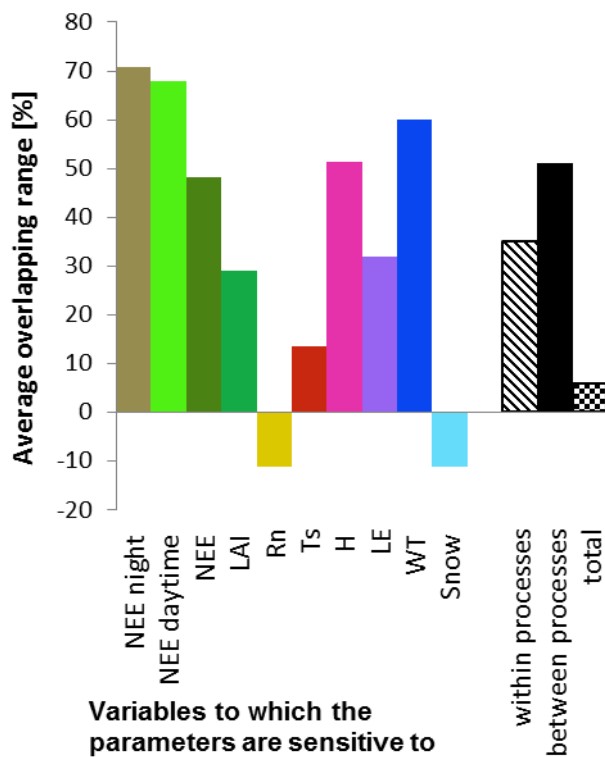

**Figure 4. Average overlap of accepted ranges per parameter within each process and between processes, i.e. how unambigously the parameters could be constrained. Negative values indicate the distance between accepted ranges when ranges did not overlap at all.**





**Figure 5. Correlations between performance indices in the prior distribution (3200 random runs): $R^2$ versus $R^2$ (upper panel); mean error (ME) versus ME (lower panel). Each of the dots represents a parameter set. Grey lines indicate the axes through zero.**



**Figure 6. Correlations between performance indices in the prior distribution (3200 random runs): $R^2$ (columns) versus mean error (ME) (rows). Each of the dots represents a parameter set. Grey lines indicate the axes through zero.**



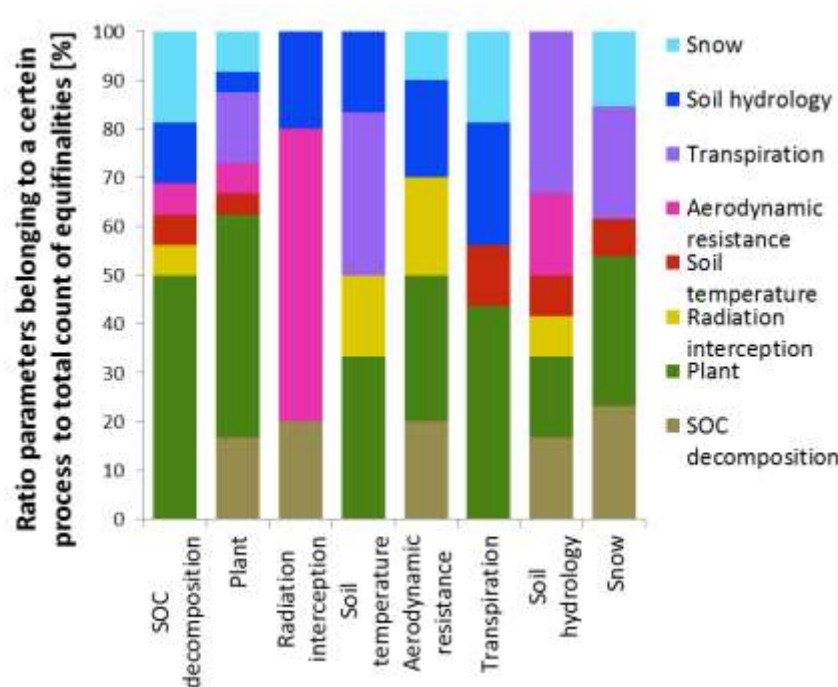

Figure 7. Module belongings of parameters that correlated with parameters of a certain module.