# Peer review of "The importance of process interactions and parameter sensitivity for modelling the carbon dynamics in a natural peatland"

_Geoscientific Model Development, 2016_

## Short Comment (SC1) · 6 Jun 2016

Dear authors,

In my role as Executive editor of GMD, I would like to bring to your attention our Editorial version 1.1:

http://www.geosci-model-dev.net/8/3487/2015/gmd-8-3487-2015.html

This highlights some requirements of papers published in GMD, which is also available on the GMD website in the 'Manuscript Types' section:

http://www.geoscientific-model-development.net/submission/manuscript_types.html

[Figure]

In particular, please note that for your paper, the following requirements have not been met in the Discussions paper:

- "The main paper must give the model name and version number (or other unique identifier) in the title."

- "If the model development relates to a single model then the model name and the version number must be included in the title of the paper. If the main intention of an article is to make a general (i.e. model independent) statement about the usefulness of a new development, but the usefulness is shown with the help of one specific model, the model name and version number must be stated in the title. The title could have a form such as, "Title outlining amazing generic advance: a case study with Model XXX (version Y)"."

Please correct this in your revised submission to GMD.

Yours,

Astrid Kerkweg

---

## Referee Comment (RC1) · Anonymous Referee #1 · 8 Jul 2016

**General comments:**

The manuscript "The importance of process interactions and parameter sensitivity for modelling the carbon dynamics in a natural peatland" describes a calibration and sensitivity analysis of the CoupModel, using several variables measured on an eddy covariance test site in Sweden. Several variables, which describe carbon, energy and water fluxes, are used for calibration and a sensitivity analysis.

Overall the manuscript is not well written and difficult to follow. In wide parts corrections by a native speaker is required. However, the study is interesting and contains relevant aspects. Unfortunately, the actual presentation of the study is not convincing. Several parts are too fuzzy and too general, while other parts are too detailed. The objective is not clear and the conclusion does not provide any new information despite general knowledge about this field. I am really puzzled to rate this manuscript, as there are many concerns in almost all parts of the manuscript. However, as I see also the potential of the study, I rate it acceptable with **major revisions**, but I have to be clear, that only answering the comments below won't be enough to get the publication to an acceptable form. As there were too many issues, it was not possible to comment all in detail, but I tried to explain my concerns on some parts in more detail.

**Overall comments:**

The objective is not really clear. Reading the paper it seems like that all variables are needed to improve the quality of calibration, several parameters are interacting and more measurements are needed. I do not need a study to come to this conclusion. I miss out more numbers, that rate the quality, and real values, like how much quality do I miss out, if I calibrate only on one variable rather than on all variables (for R2, ME and NSE). There is no discussion about transferability of the results and the robustness of the results. Which results can be used in general and which are related to the CoupModel. I do not see the list of other studies as a discussion of transferability. If the authors want to include these studies, there need to be an analysis of the differences for the different approaches used for the different processes.

I am also not sure if the picked indicators describing the goodness of fit are well picked. The mean error will compensate strong negative and positive disagreements in the overall value, which do not reflect the quality of the model performance. I would like to see the root mean square error used instead. Also R2 is not a good value for model performance as it might be sensitive to extreme values, if not all parts of the data range are represented equally.

The state of the art method for calibration is Bayesian calibration, which is not mentioned in this study. At least in the introduction and maybe in the discussion this needs to be mentioned and explained why the here used method is as good, better or worse than the Bayesian calibration and what are the advantages and disadvantages.

Using several variables for calibration of models or a sensitivity analysis the pareto optimization would be an appropriate multi-criteria approach to address the subjective judgement of the model performance. However, at least this technique and/or other approaches for multi-criteria optimizations should be mentioned and discussed.

**Specific comments:**

**Comments on the title:** First, the model used in this study is not mentioned in the title. Second, the title contains only the carbon dynamics, while most of the variables that are considered in the analysis are energy and water fluxes. The title needs to be re-formulated and more precise.

**Comments on the objective:**

- For point 1. the authors do not identify processes, but parameters and variables, which are most sensitive in the model to simulate the target variables
- Point 2. is not well formulated and it is difficult to understand the objective.
- I am not sure about point 4. Why do you test the usability of measurements? The measured data are usable. Additional, it is not true that you can detect the missing measurement variables. You can detect model sensitivity and required data for the used version of CoupModel. Other models might need other parameter sets and might show different sensitivities. Also, the authors work out improved model performance by adding more variables in the calibration process, it still rise the question, if additional measurements are need, the model approach that simulates the process needs to be improved or the calibration approach needs to be improved.
- In the objectives it is not mentioned what the authors are actually doing. The model is not mentioned and the four points are not linked to any land use, model or analysis approach.
- The sentences after point 4 do not contain any useful information about the actual study, but only general information what you can do with an outcome from a sensitivity analysis.

**Comments on the method section:**

Section 2.2: The gap-filling of the climate data is explained, but not the gap filling of the EC data. The model description is far too long, but leaves crucial aspects out at the same time. There is also lack the scientific terminology.

Page 4 line 20: EC is not defined. Please add this on line 15 the same page.

Page 4 lne 28-29: C uptake by the ecosystem from the atmosphere

Section 2.3.3 There is no need to give a general introduction into soil hydrology.

Section 2.3.4 There is no need for a general introduction into phenological models, but provide the key information: used phenological model, the model is based on temperature sum and day length, parameters and settings. Also the description of allocation of carbon in the plant is too long and not well formulated. Especially, the labelling of parameters in the model do not contribute to a better understanding of the study.

Section 2.3.5 The section is too long, in some parts the scientific terminology is not used, the description of the processes is too casual and essential information is missing (e.g. turnover rates of the pools, decomposition follows first order kinetics). Based on the description I am not sure, if the model considers really different temperature sensitivities for fungi and bacteria (which would surprise me) and where the data about community size are coming from. I assume the authors mean that the SOC module contains a parameter that controls the impact of temperature on the decomposition rates and this factor was calibrated and tested for fungi and bacteria dominated soils. For the here presented study this doesn't matter.

Section 2.4.3 A couple of problems with the NEE values could be sorted by using the correction approach by Papale et al., 2006 (Biogeosciences, 3, 571–583). This would enable to solve the problems with extreme day values and the peaks for the night periods. I also wondering, if the gap filling tool, develop by Reichstein and Falge, is used to fill gaps for NEE?

**Comments on the result section**

Page12 lines 11-14 I understand that the soil water content is an important variable, which is difficult to measure and to simulate. This is not new and as this is known, this should be a central part of a sensitivity analysis. I think it is not enough to ask for more measurements, which is always a good answer to all problems with simulations. First, I miss a discussion of the measurements of the soil water content, which is often done on a single spot rather than spatial distributed or in different depths. Second, there is no discussion of the footprint area of the EC measurements. If the footprint changes and the soil type or hydraulic properties differ on the test site, this might explain differences. Third, as the authors make a sensitivity analysis, it is possible to detect the most sensitive soil property and give at least the advice, which soil property should be measured to get better results with the CoupModel.

3.1 Parameter sensitivity:

I do not understand why the authors highlight the module dependency so strong. This analysis makes the study extremely model dependent. I think the authors

should relate the sensitivity to processes. I assume that the modules represent separate processes, but this is not necessarily the case.

Page13 lines 27-29 R2 and ME are contradicting in their goodness of fit: Is this an indication that these are not the best indicators to detect the quality of performance?

Section 3.4 Usefullness might be not a good word to describe the measured variables.

**Comments on the discussion:**

Wide parts of the discussion are not really a discussion, but do only compare qualitative findings of the study with other studies.

4.1 Parameter sensitivity

It is correct that the detection of sensitivity of parameters enable to concentrate the calibration on the main drivers, but how robust are the findings on this test site and how transferable are the results to other ecosystems or to other climate zones? Peatland in Northern Europe is a quite specific test site, so, is it possible to transfer the results to mineral soils? How transferable are the results to Central Europe or to the Mediterranean area? It is no problem, if the results are not transferable, but at least there need to be a discussion.

Page 19 lines 3-5: "While the existence of interactions between the processes and their parameters is supposed to be less dependent on site conditions and model structure, the exact shape of the connections as well as constraint parameter ranges might strongly depend on these factors. " This might be correct as the sensitivity analysis only represents effects of the model structure. However, by applying the analysis on a specific test site, the relevance of processes depends on the climate zone, ecosystem, land use, soil type, etc. This also effects the limitations for the data range of the considered parameters and variables. The relation and interaction might be different outside this range. Therefore, I wouldn't exclude the site conditions as relevant factors.

Page19 lines 14-16: It depends: Several models using the same approaches to describe processes. Therefore, the formulated hypothesis needs to be tested by compare the approaches used in the different models to be sure, that this correlations are really independent of the model structure.

Page 19 line 27 to page 20 line 2: I do not really understand how the implementation of open water bodies should explain the differences in the correlations. In the measurements H is more related to temperature and LE more to the water flows. Photosynthesis is the main driver for growth and photosynthesis is calculated by a light use efficiency function and, as written in this manuscript "….total plant growth is proportional to the net global radiation absorbed…..". Is it possible that the correlation of H and NEE can be explained by the calculation of photosynthesis by radiation, which is also the main driver for H, while LE is calculated in more complex equations with less direct correlation to radiation and temperature?

Page 20 lines 3-5: No, not necessarily. If you try to understand the pattern of data in advance, the used indicator for the goodness of fit can be picked sensible. E.g. there are variables with several values (e.g. night values) at zero or around zero. These values will have a strong impact on the ME as the models, usually, simulate the zero values during night quite well. The R2 can cope with the clouds around zero, but it is sensitive to single extreme values. Bottom line the used indicator for goodness of fit influences the outcome of the analysis and if the indicator is well picked, there are subjective judgements. Controversial results of different indicators need to be analysed to understand the reasons for the contradiction. Unfortunately, this analysis is missing in this manuscript.

Page 20 Lines 6 -15: Of course there are lot's of correlation between LAI and other variables, because these parameters use LAI. However, an analysis and discussion of the cited publications is missing. This would be a chance to bring the here presented study in the context of other studies. Instead of only mention the correlation, the authors could explain the different dependencies. E.g. I assume that LAI correlates with soil water content, if it is a dry, water limited ecosystem.

Page 20 line 17 temporal or spatial resolution? What means high resolution mm, cm or m; seconds, hours, days?

Page 23 Lines 10 – 15: I see the strong sensitivity of the soil hydraulic properties as relevant factor, but first, it is not that easy to measure these parameters and, second, I think the authors should provide an alternative method to derive better fits and quantify the reduction of quality by missing out soil hydraulic properties. An alternative method would be to calculate the soil hydraulic properties by pedo-transfer-functions (as mentioned in the model description). If do so, the sensitivity of single parameters (soil type, bulk density, field capacity (by itself) etc.) can be tested and it might be possible to get better calibration using this information or detect the most sensitive of these parameters.

**Comments on the references**

The publication of He et al. needs to be updated

**Comments on figures:**

- I would like to see a figure like Fig.5 also for actual values and not only for a prior and posterior comparison.
- The quality of the figures is not good

**Comments on the supplement:**

Table S1 I think there is no need to present parameter name in the model. I am even not sure if the module name provides any useful or needed information, but it might

be better to group the parameters instead (e.g. soil, hydrology, snow, vegetation/growth).

Table S2 is really needed, if you develop a model and publish it, but I do not see the use for the actual study. Most of the equations are standard approaches that are already described in the text.

---

## Referee Comment (RC2) · Anonymous Referee #2 · 5 Aug 2016

Metzger et al present an interesting study addressing process interactions and parameter sensitivity for model carbon dynamics in a natural peatland. This is a "heavy" topic and the authors did a good job. Their findings are important and meaningful for both model users and model developer, the latter of whom they overlooked. There are some aspects needs substantial revision. a) There are too many small paragraphs with only one or two sentences. I would suggest the authors to combine them. b) The authors claimed "interactions between parameters" "limited transferability of parameter values between models and even between studies". I am not quite understand the connections between the two topics. It could be great if the authors can elaborate more on this. c) The authors mentioned many times of "$CO_2$ model(s)", which seems improper

because the Coupmodel is more like a C cycling model, rather than $CO_2$ model. d) This work is not only meaningful for model users, but also for model developers. Nowadays, for example, many researchers develop and use models to predict impacts of climate change on carbon cycling or hydrology, and others. However, many of these models are not integrated or balanced enough representing all aspects (processes/modules). Such model predictions lack of credit for me. I could suggest the authors also discuss this aspect in the discussion section. Overall, I think the paper is publishable after major revision. Some specific comments are: 1) Line 9-10: From my understanding, most previous models focused only one or few modules because their model emphasized only on these module(s) and simplified (overlook) others. Interestingly, this could highlights the importance of the present study. The authors may want to elaborate this point more. 2) Line 13: Please specify the modules to make the reader to easy understand. 3) Line 20: This sentence is hard to understand. Please revise. 4) The introduction contains too many paragraphs and they are not very well logically connected. Please consider to reduce them into 4-5 paragraphs. 5) Line 28: I think these findings will be of critical importance for model development as well. 6) Line 1 in Page 9: What do you mean of "uniform random distribution"? 7) Line 9 in page 9: Has this definition of sensitivity been used by others? 8) Line 21 in page 9: Please explain clearer how the equifinalities was quantified. Figures quality/resolution are low. It is hard to read these figures

---

## Author Comment (AC1) · 9 Oct 2016

Thank you for the hint. Model name and version number will be included in the revised version of the manuscript.

---

## Author Comment (AC2) · 9 Oct 2016

Dear Authors, this is an interesting study, but there are still a lot of issues in the presentation of the study as well as with analysis and discussion. The application of one model on an single test site is quite specific, which makes it more important to distribute between site and model specific result and general findings. Especially the later ones I would like to see worked out and highlighted more. Please find my more detailed comments in the supplement.

Please also note the supplement to this comment:

http://www.geosci-model-dev-discuss.net/gmd-2016-116/gmd-2016-116-RC1-

supplement.pdf

General comments:

The manuscript "The importance of process interactions and parameter sensitivity for modelling the carbon dynamics in a natural peatland" describes a calibration and  sensitivity analysis of the CoupModel, using several variables measured on an eddy  covariance test site in Sweden. Several variables, which describe carbon, energy  and water fluxes, are used for calibration and a sensitivity analysis.

Overall the manuscript is not well written and difficult to follow. In wide parts corrections by a native speaker is required.

*English copy editing will be provided by the journal in a later state of the manuscript processing*

However, the study is interesting and contains relevant aspects. Unfortunately, the actual presentation of the study is not convincing. Several parts are too fuzzy and too general, while other parts are too detailed. The objective is not clear and the conclusion does not provide any new information despite general knowledge about  this field. I am really puzzled to rate this manuscript, as there are many concerns in almost all parts of the manuscript.

However, as I see also the potential of the study, I rate it acceptable with  major  revisions, but I have to be clear, that only answering the comments below won't be enough to get the publication  to an acceptable form. As there were too many issues,  it was not possible to comment all in detail, but I tried to explain my concerns on some parts in more detail.

**Overall comments:**

The objective is not really clear. Reading the paper it seems like that all variables are needed to improve the quality of calibration, several parameters are interacting and  more measurements are needed. I do not need a study to come to this conclusion.

*The main message of the study is that parameters are interlinked, not only within, but also between different modules. It implies that parameter ranges might not be transferrable between studies that use different models or even same models with a different set of calibrated parameters or included processes. This hasn't been shown before, as previous C-cycle studies on peatlands usually calibrate only parameters of the Carbon module, or from*

*few additional modules. Further, while multi-criteria constrain is widely used in e.g. hydrological modelling, this technique is still hardly found in C-cycle modelling studies, especially on peatlands. Instead, it is common practice that parameter values are transferred between studies and between models without questioning the covariance between parameters and dependence on the variable and criteria used to reject not acceptable performance of the model.*

***In the revised version, we reformulated the objectives to emphasize the understanding of the dependencies between parameter distributions and between parameters and model performance. In the results we added a sentence, telling that using several measurement variables helps to identify if a parameter range is not robust. Also in the discussion and conclusions we reformulated some parts to emphasize the importance of using many variables and criteria to constrain the model.***

I miss out more numbers, that rate the quality, and real values, like how much quality do I miss out, if I calibrate only on one variable rather than on all variables (for R2, ME and NSE).

*Calibrating using only one observation variable and criteria will normally create the highest performance for that particular variable and the particular index. However, the result can easily be unique for only that particular variable and time period used and lead to worse performance in other variables or if other indices are used. The advantage of using several variables and indices is to be able to identify which of the resulting parameter ranges vary depending on the chosen criteria and which are robust in this respect (still, this doesn't include the robustness in respect to transferability between models and sites) but is difficult to quantify. Figure 4 and 5 show how much the performance in a certain variable is reduced, if criteria for another variable or performance index is set. It would be possible to create such figures for all combinations of multiple criteria, but this would be several pages of figures.*

*If only some few parameters are calibrated, the same or similar goodness of fit might be achieved, depending on which parameters are chosen, but parameters will be constraint to a range, which may be misleading because of the tendency of equifinality. To identify the correlation structure between parameters we have to define a list of parameters that have the change to be both correlated to other parameters and sensitive to the criteria and data available.. Fig. 3 tells how many parameters can be constrained depending on which variables are used to constrain the model in the calibration procedure.*

*Out of 27 sensitive parameters, 15 could not be constrained to an unambiguous range. This means, that in more than 50% of the cases, a parameter range constrained by only one variable or index is not robust because it depends on the chosen criteria. The more variables and performance indices are used and the more parameters are calibrated, the more a statement is possible if the resulted parameter range is robust or not and to which factors it is connected to.* ***This number was added to the results and the meaning added to the discussion.***

There is no discussion about transferability of the results and the robustness of the results. Which results can be used in general and which are related to the CoupModel. I do not see the list of other studies as a discussion of transferability. If the authors want to include these

studies, there need to be an analysis of the differences for the different approaches used for the different processes.

*Interactions, also between different modules, certainly exist on other sites/ecosystems and with other models as well. The same applies to the problem of different resulting ranges depending on performance index, measurement variable and it's sub period used for calibration.*
*But the specific results, i.e. which parameter interact in which way, the constrained parameter ranges, the rank of parameter uncertainty and therefore importance of additional needed measurement variables, and the parameters identified as most sensitive are probably to a large extend model, ecosystem and maybe site specific. As we tested only one model on one site, the only way to make a statement about transferability is to compare with other studies. We mention the model name and the ecosystem of these studies, but analyzing all differences between the studies would include differences between models (used equations, processes that are implemented or not, ...), between applied methods (calibration procedure, selection of other parameters that are calibrated simultaneously, performance indices, calibration variable, tested value range of the parameter, ...) and between sites (ecosystem, climate zone, soil conditions, vegetation, ... ) - all of them might play an important role why this parameter was found to be most sensitive. A full list of the differences would just be too long, especially as this is not the main message of the study. All studies differ from ours in at least one point (e.g. ecosystem type, which is already mentioned in the manuscript), indicating that some results might be transferrable to some extent*
***We added at several positions in the discussion the information about whether a certain result relates to CoupModel and site conditions or can be used more general.***

I am also not sure if the picked indicators describing the goodness of fit are well  picked. The mean error will compensate strong negative and positive disagreements  in the overall value, which do not reflect the quality of the model performance. I  would like to see the root mean square error used instead. Also $R^2$ is not a good  value for model performance as it  might be sensitive to extreme values, if not all  parts of the data range are represented equally.

*It is true  that strong negative and positive disagreements are compensated in mean error, but they are reflected in  the $R^2$. The root mean square error has the disadvantage that is doesn't tell if there is an over- or underestimation. There are many other performance indicators, some of them calculated on base of $R^2$ or ME or the combination of both. We chose $R^2$ and ME because they are simple and we think they are sufficient to show the main message. We agree that a single performance indicator should be easier. The reason for selection both $R^2$ and ME was that we would like to distinguish errors related to the mean bias and the ability to reflect the variability in itself.  The ability to reflect the full range between high and low values and being sensitive to the magnitude of the range was part of conceptual thinking behind the criteria chosen.*

The state of the art method for calibration is Bayesian calibration, which is not  mentioned in this study. At least in the introduction and maybe in the discussion this needs to be mentioned and explained why the here used method is as good, better or worse than the Bayesian calibration and what are the advantages and  disadvantages. Using several variables for

calibration of models or a sensitivity analysis the pareto optimization would be an appropriate multi-criteria approach to address the subjective judgement of the model performance. However, at least this technique and/or other approaches for multi-criteria optimizations should be mentioned and discussed.

*The Bayesian approach have a lot of advantages providing that we have a well defined error model and that the multiple variables can be combined into one single log-likelihood value. The high number of different variables and especially the risk for converting into posterior distribution without covering the full range of combinations for all parameters was the main reason for not selection the Bayesian approach. The Bayesian approach does not show any substantial advantages when we have many different measurement variables and we would like to have an unbiased investigation of all parameter combination rather than searching for a singly highest probability of the entire model.*

**We mention the Bayesian approach in the revised version in the introduction and the discussion.**

**Specific comments:**

**Comments on the title**:

First, the model used in this study is not mentioned in the title.

**OK, done in the revised version**

Second, the title contains only the carbon dynamics, while most of the variables that are considered in the analysis are energy and water fluxes. The title needs to be reformulated and more precise.

**We reformulated the title to include also heat and water fluxes.**

**Comments on the objective:**

- For point 1. the authors do not identify processes, but parameters and variables, which are most sensitive in the model to simulate the target variables.

*Processes are described by equations, containing parameters. Parameters determine if an equation results in a high or low value. If the result of an equation doesn't matter for the fit of the model output to a variable, it means that the underlying process doesn't play an important role for the variable in the tested scenario. Therefore, identifying the most sensitive parameters means identifying the sensitive processes. An exception would be if several parameters of the same equation would be calibrated, but this was avoided in this study.*

**We reformulated this objective to make it more clear and added the explanation to the discussion.**

-Point 2. is not well formulated and it is difficult to understand the objective.

*Reformulated in the revised version.*

- I am not sure about point 4. Why do you test the usability of measurements?  The measured data are usable.

*Translation error. Should be usefulness or potential. **Replaced in the revised version.***

Additional, it is not true that you can detect  the missing measurement variables. You can detect model sensitivity and  required data for the used version of CoupModel. Other models might need  other parameter sets and might show different sensitivities.  Also, the authors work out improved model performance by adding more variables in the calibration process, it still rise the question, if additional measurements are  need, the model approach that simulates the process needs to be improved or  the calibration approach needs to be improved.

*We identify parameter that are highly sensitive and at the same time not constrainable with the available data. As we calibrate only one parameter per equation, it means that the process described by this equation plays an important role. If it is possible to measure a variable that describe this process, we found a "missing variable". E.g. the high concern of a parameter describing the soil water retention curve. This is used for calculation of the soil water content. So either having measured soil water retention, or soil water content, would improve the modeling. However the improvement is not in a better model performance, but in the possibility to constrain this and connected parameters to a more narrow range (and therefore improve predictions which might be performed with this model). Of course we tested it only for CoupModel, but it is probably also an important variable for other models that have some dependence of decomposition or plant growth from soil water content. Only for models that do not have this dependency it indicates that including such a dependency/adding corresponding processes might improve the model performance.*

*A much larger limitation might be  the dependence on site conditions. E.g. we know from measurements and correlation analyses that water level does not play an important role at every peatland, which might indicate that also water content might not play an equally important role on all peatland sites - e.g. because the water content is not much fluctuating. However, for natural peatlands with hydrological regime related to climate there are strong reasons to believe that our results are general.*

***We added "by identifying sensitive or interacting parameters that cannot be constrained by the available data" to this objective and incorporated the response to this comment in the discussion.***

- In the objectives it is not mentioned what the authors are actually doing. The  model is not mentioned and the four points are not linked to any land use,  model or analysis approach.

***We reformulated the objectives to be  more precise and added a sentence about what we are doing.***

- The sentences after point 4 do not contain any useful information about the  actual study, but only general information what you can do with an outcome  from a sensitivity analysis.

*You are right, that this information applies to sensitivity analyses in general, but this information might be still be valuable for readers, that are not very familiar with sensitivity analyses.*

**We restricted the sentence after point 4 to Carbon models and peatlands.**

**Comments on the method section :**

Section 2.2: The gap - filling of the climate data is explained, but not the gap filling of the EC data .

*The EC data was not gap-filled, as mentioned in the last sentence of Section 2.2. Only measured data were used for calibrating the model.*

The model description is far too long, but leaves crucial aspects out at the same time. There is also lack the scientific terminology.

*See response to specific comments below*

Page 4 line 20: EC is not defined. Please add this on line 15 the same page.

**OK**

Page 4 lne 28-29: C uptake by the ecosystem from the atmosphere
**OK**

Section 2.3.3 There is no need to give a general introduction into soil hydrology.
*This section describes how soil hydrology is realized in the CoupModel in the used setup. CoupModel provides many possibilities for the user to select between different sub models, different equations and different complexities of the used equations. E.g. ground water flow as well as evaporation can be included or discarded and there is no need for using the Richards equation or simulating soil water vapor in CoupModel. The number of hydrological soil horizons is flexible; instead of Brooks & Corey, the van Genuchten equation can be used for description of the water retention curve, etc. This is all configured by switches through the user - the text describes how these switches were set, which is relevant information that cannot be found in the manual.*
**We added a sentence in Section 2.3. to make it clearer, that the following sections describe the applied, study specific configuration.**

Section 2.3.4 There is no need for a general introduction into phenological models, but provide the key information: used phenological model, the model is based on temperature sum and day length, parameters and settings. Also the description of allocation of carbon in the plant is too long and not well formulated. Especially, the labelling of parameters in the model do not contribute to a better understanding of the study.
*Also for vegetation, CoupModel provides a wide range of opportunities. Mosses as additional plant layer had never been simulated before with the CoupModel, which makes it necessary to describe how the existing C pool scheme in the model was applied to mosses, that do not have roots and a seasonality comparable to vascular plants. Also for vascular plants, the carbon pools were used in an unconventional way that allows considering stems as*

*photosynthetically active and that allows senescence to be dependent on both, temperature sum and growth stage. These are also the reasons for the labeling, explaining how the model was configured and how to understand parameters and equations, that still use a labeling that was originally intended for vascular plants, in particular trees. This information is relevant for reproducing the study and of interest for other CoupModel users that would like to apply the model on a moss/sedge dominated site. **But we agree, that this section is very long and therefore moved large parts to the supplementary material.***

Section 2.3.5 The section is too long, in some parts the scientific terminology is not used, the description of the processes is too casual and essential information is missing (e.g. turnover rates of the pools, decomposition follows first order kinetics).
*Turnover rates of pools were calibrated parameters. For a better readability we did not mention any values of fixed parameter as well as value ranges of calibrated ones in the text. Instead they can be found in tables S2 and S3 in the supplement as mentioned in section 2.3, last sentence. A quite large part of this section is occupied by the description of how peat growth was simulated. This functionality was newly developed for the site in this study and therefore not described anywhere else.*

***We added the information about first order kinetics.***

Based on the description I am not sure, if the model considers really different temperature sensitivities for fungi and bacteria (which would surprise me) and where the data about community size are coming from. I assume the authors mean that the SOC module contains a parameter that controls the impact of temperature on the decomposition rates and this factor was calibrated and tested for fungi and bacteria dominated soils. For the here presented study this doesn't matter.
*There is no difference between bacteria and fungi in the used setup of the model. The bacteria and fungi in the text refers to the applicability of the Ratkowsky function that was used to describe the temperature dependence. This function was originally developed for bacteria, we mentioned the previous application to fungi by others, to justify why we applied it on a peatland, where fungal decomposition plays an important role. Other peatland models and previous CoupModel applications often use a more simple Q10 approach to describe temperature dependence.*
***In the revised manuscript we reformulated this sentence and moved the description of the temperature dependence more to the top of the section.***

Section 2.4.3 A couple of problems with the NEE values could be sorted by using the correction approach by Papale et al., 2006 (Biogeosciences, 3, 571–583). This would enable to solve the problems with extreme day values and the peaks for the night periods.
*We did not apply a filter for friction velocity or any spike removal as suggested in Papale et al 2006 for following reasons: We did not see any effect of friction velocity on the fluxes which is likely due the EC measurements being conducted at 2m above an open mire surface. Furthermore, friction velocity filtering is only valid if the turbulent transport and biological sources of measured fluxes are coupled. During nighttime, however, biological activity might continue while the turbulent transport is absent leading to accumulation of e.g. $CO_2$. The accumulated concentrations might be released and detected as 'spikes' in the morning when turbulent movement sets in. Removing these spikes would therefore introduce an error in the C budget (i.e. in the emission component). In addition, inherent noise in EC data leads to*

*occasional spikes which are presumably randomly and evenly distributed around the mean. Selectively removing spikes might introduce artificial and subjective bias into the flux balance.* **We have clarified and rephrased the relevant text in section 2.4.2.**

I also wondering, if the gap filling tool, develop by Reichstein and Falge, is used to fill gaps for NEE?

*Although gapfilled data was available based on the Reichstein et al 2005 approach, this study only used the measured NEE, H and LE fluxes and omitted all gapfilled periods, as stated in section 2.2. The model should be calibrated with measured data, not with another model.*

**Comments on the result section**

Page12 lines 11-14 I understand that the soil water content is an important variable, which is difficult to measure and to simulate. This is not new and as this is known, this should be a central part of a sensitivity analysis. I think it is not enough to ask for more measurements, which is always a good answer to all problems with simulations. First, I miss a discussion of the measurements of the soil water content, which is often done on a single spot rather than spatial distributed or in different depths. Second, there is no discussion of the footprint area of the EC measurements. If the footprint changes and the soil type or hydraulic properties differ on the test site, this might explain differences.

*The referee is correct in that continuous measurements of the water table depth is conducted at just one spot and spatially distributed measurements would give a measure of the variation. However, the mire surface within the entire footprint is totally (100%) covered by Sphagnum mosses. The most important functional trait separating Sphagnum mosses into different functional groups is the architecture of the plant determining both the capillary forces as well as the water holding capacity and thus at what distance to the water table the different Sphagnum species grow. The plant community distribution within the foot print areas (see below) is very homogenous and totally dominated by Sphagnum species (see below) reflecting a growing season average water table of ~5-15 cm below the moss surface. Thus, even if we have conducted continuous measurements of both water table at one spot and soil water content at a few spots the dominating Sphagnum species composition within the footprint clearly reflects a spatially average water table equal to the measurement spot.*
*The position of the EC tower is in the center of a mire unit totally dominated by lawns, i.e. the growing season average water table varies between ~8-15 cm below the mire surface (see e.g. Sagerfors et al 2008). The lawn plant communities have a close to 100% cover of Sphagnum mosses (Sphagnum balticum, Sphagnum majus and Sphagnum lindbergii) and a limited contribution of vascular plants, totally dominated by the sedges Eriophorum vaginatum and Trichophorum cespitosum and the dwarf shrubs Vaccinium oxycoccus and Andromeda polifolia.*
*Both the day time and night time foot prints are well within this very homogenous lawn dominated unit of the mire (see Sagerfors et al 2008). The footprint areas are most narrow with daily average 90%-tile boundaries <<50m radius to the tower with most limited seasonal variation (seasonal footprint modelled by Kljun, unpublished).*
*The need for measuring water content on several spots and in several depths is already mentioned in the manuscript: "Thereby, the horizontal and vertical variability in peat hydraulic properties needs to be accounted for (Baird et al., 2012, Waddington et al., 2015)."*
***We added the footprint area problematic to the discussion: "measured NEE is not the CO2 exchange between biosphere and atmosphere at a certain point, but is a calculation based on turbulent vertical fluxes measured several meters above the ground and resulting from a***

***diurnal and seasonally changing area that includes different soil conditions and vegetation."***

Third, as the authors make a sensitivity analysis, it is possible to detect the most sensitive soil property and give at least the advice, which soil property should be measured to get better results with the CoupModel.

*We advise to measure the water retention curve. This is mentioned in the text. As we calibrated only one parameter of this curve to avoid equifinalities within the same equation, the sensitivity to this parameter represents the sensitivity to the result of the equation. **This was added in the revised version to the discussion section 4.5.***

3.1 Parameter sensitivity:
I do not understand why the authors highlight the module dependency so strong. This analysis makes the study extremely model dependent. I think the authors should relate the sensitivity to processes. I assume that the modules represent separate processes, but this is not necessarily the case.

*Processes is an ambiguous term, as it can refer to a single equation, or a set of several equations. We used module when we were talking about a process described by a set of several equations. But this seems to be ambiguous as well. **Therefore, we replaced it in the revised version by "category of processes" which we define in the beginning of the manuscript.***

Page13 lines 27-29 R2 and ME are contradicting in their goodness of fit: Is this an indication that these are not the best indicators to detect the quality of performance?

*They measure the performance in different ways: R2 measures the performance in the dynamics, whereas ME shows how well the magnitude was simulated. When they constrain a parameter to different value ranges, it means that there is no value that can produce a perfect fit in both, dynamics and magnitude. That's what we wanted to show: parameter ranges that are constrained by calibration might depend on the performance index that was chosen for calibration. R2 and ME are simple, but sufficient to show this. Of course we could compare the resulting parameter ranges with further other indices - and would get other resulting ranges. Taking only one index for calibration will give one resulting range, but does not tell the user, if there were shortcomings in either magnitude or dynamics, or something else. Reasons for the mismatching ranges is not a bad performance index but the limitation of the model to produce a perfect fit of the model output to the measured values simultaneously in both magnitude and dynamics in the certain variable. Models are always a simplification, not perfect and include parameters for which a perfect value is not existing.*

***The response to this comment is added in the revised version of the manuscript.***

Section 3.4 Usefullness might be not a good word to describe the measured variables.
*Translation error. **Replaced by potential**.*

**Comments on the discussion:**
Wide parts of the discussion are not really a discussion, but do only compare qualitative findings of the study with other studies.

4.1 Parameter sensitivity

It is correct that the detection of sensitivity of parameters enable to concentrate the calibration on the main drivers, but how robust are the findings on this test site and how transferable are the results to other ecosystems or to other climate zones? Peatland in Northern Europe is a quite specific test site, so, is it possible to transfer the results to mineral soils? How transferable are the results to Central Europe or to the Mediterranean area? It is no problem, if the results are not transferable, but at least there need to be a discussion.

*We tested only one model on one site, therefore we cannot name which of the most sensitive parameters, parameter ranges, interactions, etc. might be transferrable and also not to what extent/to which other ecosystems or models. The only indication we have, is when comparing with other studies: as mentioned in the manuscript, some parameters that we identified as most sensitive that were also among the most sensitive in studies on other ecosystem, using other models.*

Page 19 lines 3-5: "While the existence of interactions between the processes and their parameters is supposed to be less dependent on site conditions and model structure, the exact shape of the connections as well as constraint parameter ranges might strongly depend on these factors. " This might be correct as the sensitivity analysis only represents effects of the model structure. However, by applying the analysis on a specific test site, the relevance of processes depends on the climate zone, ecosystem, land use, soil type, etc. This also effects the limitations for the data range of the considered parameters and variables. The relation and interaction might be different outside this range. Therefore, I wouldn't exclude the site conditions as relevant factors.

*We fully agree, but that's more or less what we are saying.* **We didn't mention the relevance, but added it in the revised version as site and model dependent finding.**

Page19 lines 14-16: It depends: Several models using the same approaches to describe processes. Therefore, the formulated hypothesis needs to be tested by compare the approaches used in the different models to be sure, that this correlations are really independent of the model structure.

*Models often use same or similar equations, but the combination of equations, which processes are simulated and which replaced by a constant value, the number and type of parameters calibrated together and used variables for calibration differ between the studies. A detail presentation of all differences is outside the scope of this study.*

Page 19 line 27 to page 20 line 2: I do not really understand how the implementation of open water bodies should explain the differences in the correlations. In the measurements H is more related to temperature and LE more to the water flows. Photosynthesis is the main driver for growth and photosynthesis is calculated by a light use efficiency function and, as written in this manuscript "….total plant growth is proportional to the net global radiation absorbed…..". Is it possible that the correlation of H and NEE can be explained by the calculation of photosynthesis by radiation, which is also the main driver for H, while LE is calculated in more complex equations with less direct correlation to radiation and temperature?

*It is not H and NEE that correlate, but the parameter values that lead to a good fit in both. As we mention in ln 31 the same page, we tested only the effect of parameters, not the effect of input variables (like the sensitivity to radiation), which would be an interesting study as well.*

*Open water bodies is just an example for missing processes. The fit for LE is not good in spring, whereas this pattern cannot be seen in NEE. In the real world, there might be a lot of evaporation from open water bodies, so the model underestimates LE in spring - this could be compensated with parameters that lead to a higher plant transpiration (=> better fit in LE), but these parameters would also lead to an overestimation of NEE in spring (=> worse fit in NEE).* **We reformulated the sentence in the revised manuscript to make it more clear.**

Page 20 lines 3-5: No, not necessarily. If you try to understand the pattern of data in advance, the used indicator for the goodness of fit can be picked sensible. E.g. there are variables with several values (e.g. night values) at zero or around zero. These values will have a strong impact on the ME as the models, usually, simulate the zero values during night quite well. The $R^2$ can cope with the clouds around zero, but it is sensitive to single extreme values.
*To reduce the effect of extreme values, we had additionally the $R^2$ of accumulated values. As stated before, there are many more complex indices, and they would probably result in different parameter ranges - this only supports our statement: the choice of the index has an effect on the resulting range.*
*Values around zero do not have a strong impact on ME, as the modeled values during this periods are also low. That's why we decided to add ME of winter values - values are low and if you only look to the whole year, parameters that influence winter fluxes have no/low sensitivity. In case of NEE (where we differentiate between day and night values), the night values are dominated by respiration, whereas during daytime photosynthesis plays an important role - therefore it is not surprising, that different parameter and parameter ranges lead to the best fit for either day or night. This cannot be solved by a more sophisticated performance index - the underlying problem is, that the model is not able to give simultaneously a very good fit in daytime, nighttime, as well as in magnitude and dynamics - it remains a decision of the user to calibrate to NEE only, or separately to night and daytime values and to decide if a good model result in magnitude or in dynamics is more important. Same for the seasonality - if none of the runs shows the best fit in both spring and summer, it is not a question of another performance index - instead it hints to limitations in the model, e.g. a process that is not implemented or at least not included in the calibration. But models are never perfect, therefore a best value or value range is not existing for many parameters*
**We added some discussion on this in section 4.2 in the revised version**

Bottom line the used indicator for goodness of fit influences the outcome of the analysis and if the indicator is well picked, there are subjective judgements. Controversial results of different indicators need to be analysed to understand the reasons for the contradiction. Unfortunately, this analysis is missing in this manuscript.
*The most pronounced controversial results are analysed in the subsections of 4.5., but a detailed analysis for each parameter and each variable would be extremely extensive and outside the scope of the manuscript.*

**Possible reasons for controversial results, which we added now to section 4.2.:**
*- Most important: The model does not reflect the real world (e.g. decomposition rate coefficient is not a constant value, but depends on the activity of soil microbes which is influenced by many factors that vary in time, e.g. community structure, community size, stress factors, food availability and quality, etc). A parameter with very high discrepancy between performance indices is the aerodynamic resistance dependency on LAI $r_{alai}$. For a good magnitude of temperature, this value has to be extraordinary high - much higher than a value that was actually measured at this site, see discussion to this in 4.5.3.*

*- Measurements do not reflect the real world. Measurements have limitations, e.g. NEE is not the real exchange between Eddy fetch - not a point like the model, but instead an area, and the area changes during the day and during the year. Also, not the CO2 exchange between biosphere and atmosphere is measured, but turbulent vertical fluxes at the sensor (several meters above the ground), which further include a lot of calculations to receive NEE.*
*- Indicator not good (this is the case for snow - timing of snow melt is most important, but not well enough reflected in R2 and not at all in ME, see section 4.5.4)*

Page 20 Lines 6 -15: Of course there are lot's of correlation between LAI and other variables, because these parameters use LAI. However, an analysis and discussion of the cited publications is missing. This would be a chance to bring the here presented study in the context of other studies. Instead of only mention the correlation, the authors could explain the different dependencies. E.g. I assume that LAI correlates with soil water content, if it is a dry, water limited ecosystem.
*Also on peatlands, LAI correlates with water content due to transpiration. Such dependencies are nicely described in Schulze 2006. How they are realised by the different equations that have LAI as parameter is described in the supplementary material and the CoupModel description. To all three references, we refer in the manuscript, page 20, ln 13-15: "These relationships can be explained by the many dependencies between LAI and e.g. photosynthesis, transpiration, heat insulation and water uptake (Schulze 2006), of which several are also implemented in the model (see model description and equations, Sect. 2.3, Table S2 in the supplement and Jansson and Karlberg, 2010)."*

Page 20 line 17 temporal or spatial resolution? What means high resolution mm, cm or m; seconds, hours, days?
*It refers to temporal, **which we added in the manuscript**. "High" depends on what one is interested in. We worked with hourly values, so that it is sufficient to measure a time series of hourly measurements in one layer (for simulating the dynamics) plus - for the magnitude - theoretically one single measurement in the upper and on in the lower layer.*

Page 23 Lines 10 – 15: I see the strong sensitivity of the soil hydraulic properties as relevant factor, but first, it is not that easy to measure these parameters and, second, I think the authors should provide an alternative method to derive better fits and quantify the reduction of quality by missing out soil hydraulic properties. An alternative method would be to calculate the soil hydraulic properties by pedo-transfer-functions (as mentioned in the model description). If do so, the sensitivity of single parameters (soil type, bulk density, field capacity (by itself) etc.) can be tested and it might be possible to get better calibration using this information or detect the most sensitive of these parameters.
*This doesn't work on peatlands, only on mineral soil.*

**Comments on the references**
The publication of He et al. needs to be updated
***Done***

**Comments on figures:**
- I would like to see a figure like Fig.5 also for actual values and not only for a prior and posterior comparison.

*Plotting the dependencies between different output variables would require many dimensions, as they are all connected between each other and also depend on the different parameter sets. There is an enormous amount of combinations, which makes it not visualisable.*

- The quality of the figures is not good
***Figures will be uploaded with higher resolution in the revised version.***

**Comments on the supplement:**
Table S1 I think there is no need to present parameter name in the model.
*This information would be very helpful for other CoupModel users, as these are the names given in the user interface.*

I am even not sure if the module name provides any useful or needed information, but it might be better to group the parameters instead (e.g. soil, hydrology, snow, vegetation/growth).
*The parameters are sorted for the module which gives shortly in which calculation the parameter is used. Of course this could be also read from the equation number, but the text is easier to read.*

Table S2 is really needed, if you develop a model and publish it, but I do not see the use for the actual study. Most of the equations are standard approaches that are already described in the text.
*As mentioned before, there are many possibilities to configure CoupModel. Therefore the used equations vary between studies, and in some cases also the terms within an equation are modified, deepening on switches and parameters that might set a term to zero. It further shows where the specific parameters are used in an equation*

---

## Author Comment (AC3) · 9 Oct 2016

Metzger et al present an interesting study addressing process interactions and parameter sensitivity for model carbon dynamics in a natural peatland. This is a "heavy" topic and the authors did a good job. Their findings are important and meaningful for both model users and model developer, the latter of whom they overlooked. There are some aspects needs substantial revision.

a) There are too many small paragraphs with only one or two sentences. I would suggest the authors to combine them.

*OK, done in the revised version.*

b) The authors claimed "interactions between parameters" "limited transferability of parameter values between models and even between studies". I am not quite understand the connections between the two topics. It could be great if the authors can elaborate more on this.

*If parameters interact, the value range resulting from a calibration depends on the values of other parameters. This demonstrate that parameters are not independent. Therefore, one cannot transfer the information obtained from a single parameter without also considering the value of the correlated parameters. The correlation obtained may be a phenomena that is related to a possible coexistence for this particular ecosystem, But it can also be because of the problem to constrain the model by not enough of data. Note that all parameters for the posterior distribution are uncertain and we do not expect to find a narrow range for single parameters since also the real world system is expected to have a range of parameters that represent the certain temporal and spatial variability of the system considered*
***We reformulated this (in the first part of the discussion) to be more clear.***

c) The authors mentioned many times of "CO2 model(s)", which seems improper because the CoupModel is more like a C cycling model, rather than CO2 model.

*We agree that the study is not with emphasize on CO2 instead it will try to understand the full carbon turnover at the specific site. However, the use of NEE from flux measurements is of course the major response to all the ongoing processes and fluxes of the ecosystem.*
***We changed the title to include heat and water fluxes.***

d) This work is not only meaningful for model users, but also for model developers. Nowadays, for example, many researchers develop and use models to predict impacts of climate change on carbon cycling or hydrology, and others. However, many of these models are not integrated or balanced enough representing all aspects (processes/modules). Such model predictions lack of credit for me. I could suggest the authors also discuss this aspect in the discussion section. Overall, I think the paper is publishable after major revision. Some specific comments are:

*With "modellers" we mean not only model users but also model developers. **We included them more explicitly at several points in the revised manuscript.***

1) Line 9-10: From my understanding, most previous models focused only one or few modules because their model emphasized only on these module(s) and simplified (overlook) others. Interestingly, this could highlights the importance of the present study. The authors may want to elaborate this point more.

*Models are always a simplification, and even that we show the interactions between the different modules, we would not like to devalue simpler models - it always depends on how accurate the model prediction need to be.*
**We included the importance of considering the different processes together in the last sentence of the revised abstract.**

2) Line 13: Please specify the modules to make the reader to easy understand.

*OK, done in the revised version*

3) Line 20: This sentence is hard to understand. Please revise.

*OK, done in the revised version*

4) The introduction contains too many paragraphs and they are not very well logically connected. Please consider to reduce them into 4-5 paragraphs.

**We reordered the paragraphs in the introduction in a more logical order and combined them, including some reformulations.**

5) Line 28: I think these findings will be of critical importance for model development as well.

**We added the model development at several places in the manuscript.**

6) Line 1 in Page 9: What do you mean of "uniform random distribution"?

*The values are randomly taken, whereas all values in the range have the same probability of being used -* **this is added in the revised version**

7) Line 9 in page 9: Has this definition of sensitivity been used by others?

*There are several possibilities to quantify sensitivity. Most common are measures of the difference between prior and posterior parameter distribution. As we use a simple uniform distribution, it is not necessary to use sophisticated methods like Kolmogorov D statistic or stepwise regression analysis. The simplest way is to just compare the range of posterior and prior distribution. This has certainly be done in one or another way by other studies as well. In contrast to the R2 value between parameter values and performance, this accounts also for parameters that have an optimum range around in the centre of the prior distribution.*

8) Line 21 in page 9: Please explain clearer how the equifinalities was quantified.

**Reformulated to: "Equifinalities were quantified by the R2 value of a simple linear regression through the values of the interacting parameter pair in the accepted runs."**

Figures quality/resolution are low. It is hard to read these figures

*Higher resolution will be provided in revised version.*